# The NTP generating activity of pyruvate kinase II is critical for apicoplast maintenance in *Plasmodium falciparum*

**Russell P Swift, Krithika Rajaram, Cyrianne Keutcha, Hans B Liu, Bobby Kwan, Amanda Dziedzic, Anne E Jedlicka, Sean T Prigge***

Department of Molecular Microbiology and Immunology, Johns Hopkins Bloomberg School of Public Health, Baltimore, United States

**Abstract** The apicoplast of *Plasmodium falciparum* parasites is believed to rely on the import of three-carbon phosphate compounds for use in organelle anabolic pathways, in addition to the generation of energy and reducing power within the organelle. We generated a series of genetic deletions in an apicoplast metabolic bypass line to determine which genes involved in apicoplast carbon metabolism are required for blood-stage parasite survival and organelle maintenance. We found that pyruvate kinase II (PyrKII) is essential for organelle maintenance, but that production of pyruvate by PyrKII is not responsible for this phenomenon. Enzymatic characterization of PyrKII revealed activity against all NDPs and dNDPs tested, suggesting that it may be capable of generating a broad range of nucleotide triphosphates. Conditional mislocalization of PyrKII resulted in decreased transcript levels within the apicoplast that preceded organelle disruption, suggesting that PyrKII is required for organelle maintenance due to its role in nucleotide triphosphate generation.

**\*For correspondence:**
sprigge@jhsph.edu

**Competing interests:** The authors declare that no competing interests exist.

## Introduction

With increasing resistance to current front-line antimalarials, there is a crucial need to find new therapeutic interventions with novel mechanisms of action (*Dondorp et al., 2009*; *Trape, 2001*). The apicoplast organelle within the parasite has often been considered as a source of new drug targets since it is required for blood-stage survival, in addition to possessing evolutionarily distinct biochemical pathways that are not present in the human host (*Goodman and McFadden, 2013*; *Janouskovec et al., 2010*; *Mukherjee and Sadhukhan, 2016*). One pathway within the apicoplast that represents a source of potential drug targets is carbon metabolism, due to its predicted role in supporting metabolism within the organelle.

In order to fuel metabolism within the apicoplast, sources of reducing power, carbon backbones, and energy in the form of nucleotide triphosphates are required. It is hypothesized that the apicoplast meets these needs primarily through the import and metabolism of three-carbon phosphates generated by parasite glycolysis (*Lim et al., 2010*; *Ralph et al., 2004*; *Zocher et al., 2012*). These three-carbon phosphate compounds include dihydroxyacetone phosphate (DHAP) and phosphoenolpyruvate (PEP), which are imported into the organelle through the outer triose phosphate transporter (oTPT) and inner triose phosphate transporter (iTPT) embedded in the outer and inner apicoplast membranes, respectively (*Figure 1*; *Lim et al., 2010*; *Lim et al., 2016*; *Mullin et al., 2006*).

Within the apicoplast, DHAP is believed to be converted into glyceraldehyde 3-phosphate (GA3P) by the enzyme triose phosphate isomerase (TPI) (*Ralph et al., 2004*). GA3P and PEP are then fed into the methylerythritol phosphate (MEP) isoprenoid pathway (*Lim and McFadden, 2010*). These substrates first interact with 1-deoxy-D-xylulose-5-phosphate (DOXP) synthase (DXS) to form

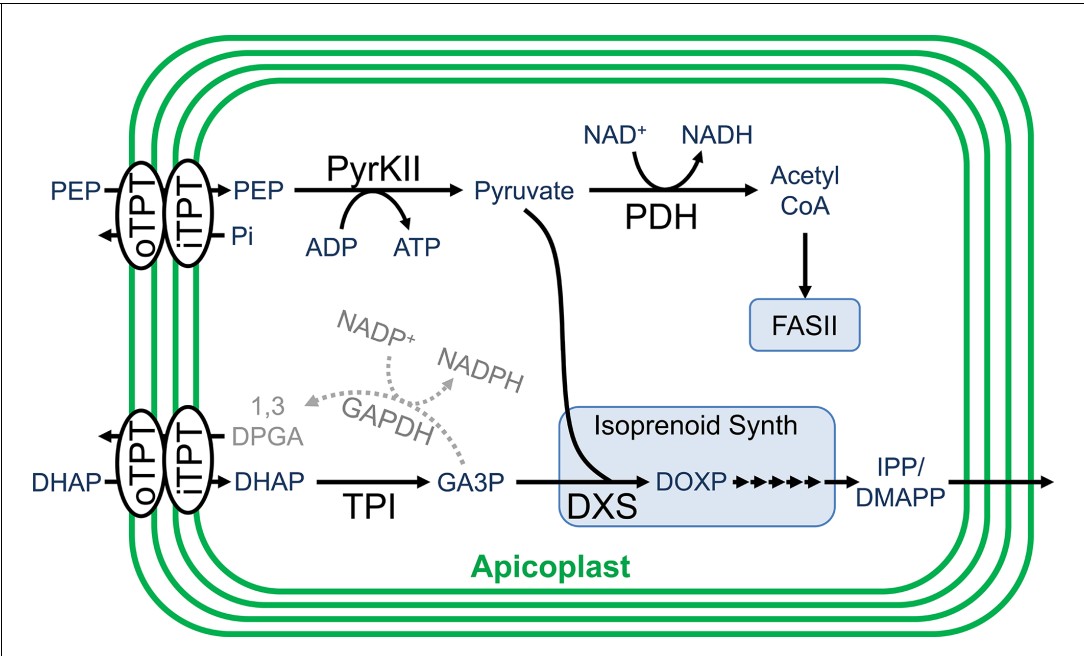

**Figure 1.** Carbon metabolism in the apicoplast. Carbon backbones are imported into the apicoplast organelle in the form of 3-carbon phosphates by the apicoplast membrane outer triose phosphate transporter (oTPT) and inner triose phosphate transporter (iTPT). Pyruvate kinase II (PyrKII) converts the substrates phophoenolpyruvate (PEP) and adenosine diphosphate (ADP) into the products pyruvate and adenosine triphosphate (ATP). Pyruvate is used by PDH to generate acetyl CoA, which is fed into the FASII fatty acid biosynthesis pathway. Triose phosphate isomerase (TPI) converts dihydroxyacetone phosphate (DHAP) into glyceraldehyde-3-phosphate (GA3P). Pyruvate and GA3P are used by 1-deoxy-D-xylulose-5-phosphate (DOXP) synthase (DXS) to form DOXP, which is further converted by a series of enzymes to form the essential isoprenoid precursors isopentenyl pyrophosphate (IPP) and dimethylallyl pyrophosphate (DMAPP). Colored in gray, a hypothetical glyceraldehyde 3-phosphate dehydrogenase (GAPDH) could convert GA3P into 1,3-diphosphoglycerate (1,3-DPGA).

DOXP, which is further converted by a series of enzymes to form the essential isoprenoid precursors isopentenyl pyrophosphate (IPP) and dimethylallyl pyrophosphate (DMAPP) (*Hunter, 2007*). GA3P is also believed to be used for the generation of reducing power through its conversion into 1,3-diphosphoglycerate (1,3-DPGA) by glyceraldehyde 3-phosphate dehydrogenase (GAPDH), reducing $NADP^+$ to NADPH in the process (*Fleige et al., 2007*; *Waller et al., 1998*). However, the functionality of the GAPDH cycle is questionable since GAPDH does not appear to localize to the apicoplast (*Daubenberger et al., 2003*).

The PEP imported into the apicoplast is believed to be used along with ADP as substrates by the enzyme pyruvate kinase II (PyrKII) to generate pyruvate and ATP, providing the only predicted source of these products within the apicoplast (*Lim et al., 2010*; *Lim and McFadden, 2010*; *Israelsen and Vander Heiden, 2015*; *Maeda et al., 2009*). The pyruvate generated enters into the MEP isoprenoid biosynthetic pathway (*Lim and McFadden, 2010*). Additionally, pyruvate can be converted into acetyl-CoA by pyruvate dehydrogenase (PDH), reducing $NAD^+$ to NADH in the process, providing one of the few predicted sources of reducing power within the organelle (*Zocher et al., 2012*; *Foth et al., 2005*). The ATP generated presumably powers all energy requiring reactions within the organelle and provides adenine bases for RNA synthesis. However, it is not clear how the apicoplast acquires the remaining NTPs for RNA synthesis and other reactions since apicoplast transporters for NMPs or NDPs have not been described, and there are no other predicted sources of NTPs within the organelle (*Ralph et al., 2004*; *Maeda et al., 2009*). The activity of PyrKII from the closely related apicomplexan *Toxoplasma gondii* has been characterized in vitro and exhibits a strong substrate preference for GDP (*Saito et al., 2008*), suggesting the likely requirement for additional NTP sources. However, the substrate specificity of the *P. falciparum* PyrKII has not been characterized, and doing so may help elucidate its potential role in apicoplast nucleotide metabolism.

In order to interrogate the role and essentiality of the various proteins involved in carbon metabolism within the organelle, we used an apicoplast metabolic bypass line to generate a series of genetic deletions. Our results showed that both TPI and PDH are dispensable for blood-stage parasite survival, and do not require mevalonate for survival. Additionally, we demonstrated that oTPT, iTPT, and PyrKII are not only essential, but are also required for apicoplast maintenance. However, deletions in the FASII (*Foth et al., 2005*; *van Schaijk et al., 2014*) and MEP pathways, the only two pathways predicted to use pyruvate, did not result in the same phenotype. This suggested that organelle maintenance is not reliant on the pyruvate generated by PyrKII, but instead may rely on ATP (or possibly other NTPs) produced by the enzyme.

In vitro enzymatic characterization of PyrKII revealed activity with all NDPs and dNDPs that were tested, demonstrating broad substrate specificity. We hypothesized that PyrKII may be playing a critical role in NTP production within the apicoplast. To test this, we generated a parasite line that allowed us to conditionally mislocalize PyrKII and measure the rate of loss of apicoplast mRNA via microarray, as a proxy for measuring the change in NTP production. Using this approach, we found that mRNA levels within the organelle decreased relatively rapidly (within ~24–40 hr of inducing PyrKII mislocalization), while the organelle did not become disrupted until much later (~88–110 hr). These results help to support the hypothesis that PyrKII plays a critical role in NTP production within the apicoplast, with the removal of PyrKII resulting in a decrease in apicoplast transcript levels, likely causing the resulting organelle loss.

## Results

### Essentiality of the outer and inner triose phosphate transporters

For the apicoplast, carbon metabolism begins with the import of the three-carbon phosphate compounds PEP and DHAP, mediated by the outer and inner triose phosphate transporters (oTPT/iTPT) (*Lim et al., 2010*). Previous deletions in *P. berghei* demonstrated that oTPT is essential, while iTPT is dispensable in blood-stage parasites (*Banerjee et al., 2012*). To investigate the role and essentiality of carbon metabolism within the organelle of *P. falciparum* parasites, we attempted to delete the genes encoding oTPT (PF3D7_0508300) and iTPT (PF3D7_0530200) using Cas9-mediated genome editing (*Ghorbal et al., 2014*; *Wagner et al., 2014*). Deletions were generated in the PfMev apicoplast bypass line, which is engineered to convert supplemented mevalonate into isoprenoid precursors in the parasite cytosol, allowing these parasites to survive disruption of the apicoplast organelle (*Swift et al., 2020*). We confirmed the deletion of oTPT and iTPT, and the absence of parental parasites using genotyping PCR reactions (*Figures 2a* and *1*). We were unable to amplify a gene from the apicoplast genome (*sufB*) in the PfMev Δ*otpt* and PfMev Δ*itpt* parasite lines, indicating that the organellar genome had been lost in both lines (*Figure 2b*; *Gisselberg et al., 2013*). To visualize the apicoplast, super-folder green (SFG) was appended to the first 55 amino acids of the *P. falciparum* ACP (api-SFG), constituting the signal and transit peptide to direct trafficking to the apicoplast, as previously described (*Swift et al., 2020*). Disruption of the apicoplast was confirmed via live epifluorescence microscopy, which showed multiple vesicles throughout the cell instead of the discrete intact structure that is typically observed in wild type parasites (*Figure 2c*; *Gisselberg et al., 2013*; *Yeh and DeRisi, 2011*). Additionally, upon removal of mevalonate both deletion lines failed to grow (*Figure 2d*). These results demonstrate that both oTPT and iTPT are required for apicoplast maintenance in addition to blood-stage parasite survival. While these results are consistent with what was found in *P. berghei* for oTPT they are notably in conflict with what was found for iTPT, which did not cause a significant growth defect in *P. berghei* when deleted (*Banerjee et al., 2012*).

### Deletion of carbon metabolizing enzymes

Once PEP and DHAP enter the apicoplast, they are metabolized by TPI and PyrKII. TPI is predicted to convert DHAP to GA3P, which is used by the MEP isoprenoid pathway (*Ralph et al., 2004*). TPI is also believed to be involved in the generation of reducing power (NADPH) by providing the substrate for a hypothetical GAPDH (*Ralph et al., 2004*). PyrKII is predicted to convert PEP and ADP into pyruvate and ATP within the organelle, providing the only predicted source of these compounds, with pyruvate being used by the downstream FASII and MEP pathways (*Ralph et al., 2004*). We were successful in deleting both TPI (PF3D7_0318800) and PyrKII (PF3D7_1037100) in PfMev

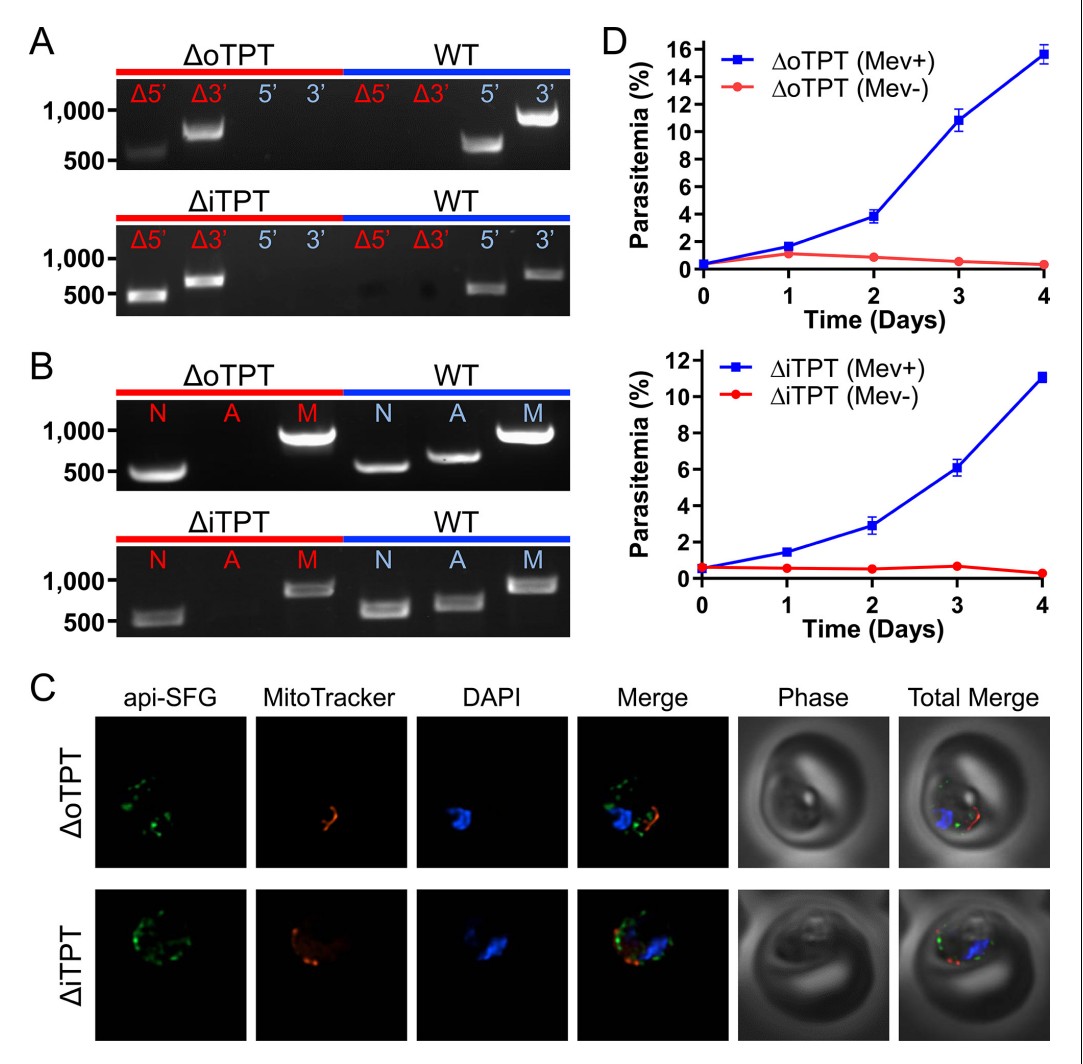

**Figure 2.** Characterization of the PfMev Δ*otpt* and Δ*itpt* parasite lines. (**A**) Genotyping PCR confirming deletion of oTPT or iTPT, with amplification demonstrating integration of the drug resistance cassette at the 5' and 3' loci of the target gene (Δ5' and Δ3'), and lack of wild type parasites due to failure to amplify at the wild type 5' and 3' loci in the PfMev Δ*otpt* or Δ*itpt* parasites, as compared to the parental control (WT). (**B**) Attempted PCR detection of the *ldh*, *sufB*, and *cox1* genes from the nuclear (N), apicoplast (A), and mitochondrial (M) genomes, respectively. We failed to amplify *sufB* from the PfMev Δ*otpt* or Δ*itpt* parasite lines, indicating loss of the apicoplast organelle genome. (**C**) Live epifluorescence microscopy of the PfMev Δ*otpt* and Δ*itpt* parasite lines, which express api-SFG (green), and are also stained with MitoTracker (red) and DAPI (blue). Microscopy images represent fields that are 10 μM long by 10 μM wide. (**D**) Growth curves of the PfMev Δ*otpt* and Δ*itpt* parasite lines cultured with or without 50 μM mevalonate. Error bars represent the standard error of the mean from two independent experiments, each conducted in quadruplicate.

The online version of this article includes the following figure supplement(s) for figure 2:

**Figure supplement 1.** Schematic and details describing gene deletion experiments.

parasites under mevalonate supplementation and confirmed the deletions by PCR (*Figure 3a*). Surprisingly, we found that the apicoplast appeared to be unaffected by the deletion of TPI. We were able to amplify the *sufB* gene using PCR, indicating that the apicoplast genome had been maintained after TPI deletion (*Figure 3b*). Consistent with this observation, the organelle appeared to have an intact morphology in epifluorescence microscopy images (*Figure 3c*). We also observed that PfMev Δ*tpi* parasites did not require mevalonate supplementation (*Figure 3d*). These results demonstrate that TPI, and its proposed activity in converting DHAP to GA3P are not required for asexual blood-stage parasite survival.

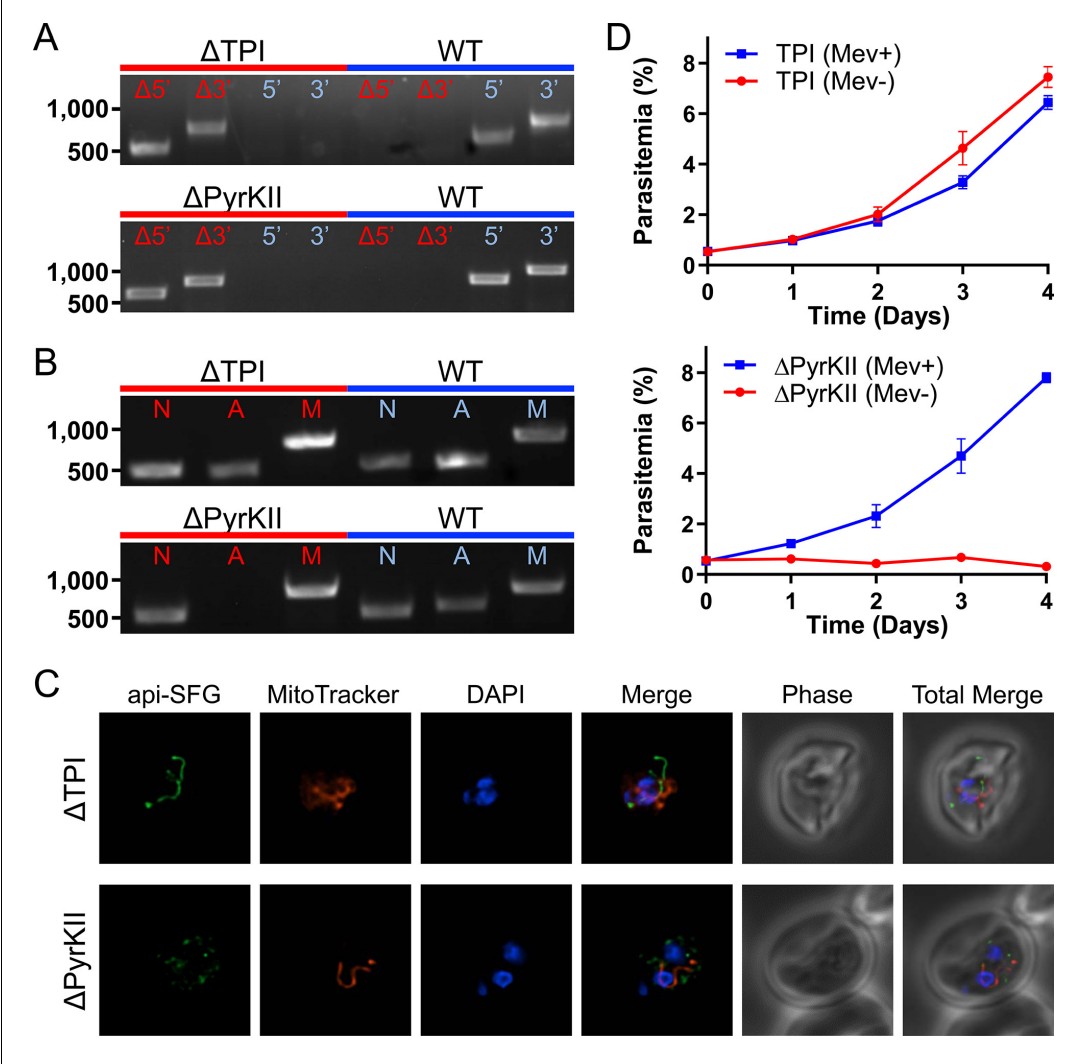

**Figure 3.** Characterization of the PfMev Δ*tpi* and Δ*pyrkII* parasite lines. (**A**) Genotyping PCR confirming deletion of TPI or PyrKII, with amplification demonstrating drug cassette integration at the 5' and 3' loci of the target gene (Δ5' and Δ3'), and lack of wild-type parasites due to failure to amplify at the wild-type 5' and 3' loci in the PfMev Δ*tpi* or Δ*pyrkII* parasite lines, as compared to the parental control (WT). (**B**) Attempted PCR detection of the *ldh*, *sufB*, and *cox1* genes from the nuclear (N), apicoplast (A), and mitochondrial (M) genomes, respectively. We were successful in amplifying *sufB* from the PfMev Δ*tpi* parasite line, indicating retention of the apicoplast organelle genome. We failed to amplify *sufB* from the PfMev Δ*pyrkII* parasite line, indicating loss of the apicoplast organelle genome. (**C**) Live epifluorescence microscopy of the PfMev Δ*tpi* and Δ*pyrkII* parasite lines. These parasite lines express api-SFG (green) and are also stained with MitoTracker (red) and DAPI (blue). Microscopy images represent fields that are 10 µM long by 10 µM wide. (**D**) Growth curves of the PfMev Δ*tpi* and Δ*pyrkII* parasite lines cultured with or without 50 µM mevalonate. Error bars represent the standard error of the mean from two independent experiments, each conducted in quadruplicate.

The deletion of PyrKII resulted in a different outcome, causing disruption of the apicoplast. This was evidenced by the failure to amplify *sufB* from the apicoplast organelle genome (*Figure 3b*) and was confirmed via epifluorescence microscopy, which showed the accumulation of multiple vesicles instead of an intact organelle (*Figure 3c*). Additionally, when we removed mevalonate from the growth medium, there was a rapid reduction in parasite growth (*Figure 3d*). These data demonstrate that PyrKII is not only essential for blood-stage parasite survival, but that it is also required for apicoplast maintenance. We also observed this phenotype for both PfMev Δ*otpt* and PfMev Δ*itpt* parasites (*Figure 2*), suggesting that apicoplast maintenance requires the import of PEP by oTPT and iTPT, likely for the generation of pyruvate and/or ATP by PyrKII.

## The role of pyruvate in the apicoplast

Pyruvate generated by PyrKII can be used by the MEP pathway to generate isoprenoid precursors (*Lim and McFadden, 2010*). Additionally, pyruvate can also be used by PDH to generate acetyl-CoA, which enters into the FASII pathway, with PDH converting $NAD^+$ to NADH in the process, potentially providing a key source of reducing power to the organelle (*Ralph et al., 2004*; *Foth et al., 2005*). In order to investigate the importance and role of pyruvate within the apicoplast, we generated deletions within the pathways that use pyruvate as a substrate. Specifically, we targeted the PDH E2 subunit of the PDH complex, in addition to the first enzyme of the MEP pathway, DXS.

We were successful in deleting the PDH E2 subunit (PF3D7_1020800) as well as DXS (PF3D7_1337200) in PfMev parasites under mevalonate supplementation and confirmed the deletions by PCR (*Figure 4a*). We also characterized the resultant apicoplast phenotypes in both of the deletion lines, using the previously described apicoplast genome PCR and epifluorescence microscopy techniques (*Figure 4b and c*). In both of the deletion lines, the apicoplast organelles appeared to be intact, indicating that these proteins are not required for apicoplast maintenance. We determined the essentiality of these proteins by testing whether the deletion lines could survive without the mevalonate supplement. We found that the PDH E2 deletion line grows equally well in the presence or absence of mevalonate, showing that PDH E2 is dispensable in blood-stage parasites. However, we found that PfMev Δ*dxs* parasites are mevalonate-dependent, demonstrating that DXS is an essential enzyme (*Figure 4d*). Taken together, these results show that the enzymes predicted to use pyruvate can be deleted without causing the loss of the apicoplast, suggesting that the product of PyrKII that is required for organelle maintenance is likely ATP rather than pyruvate.

## Enzymatic characterization of pure recombinant *P. falciparum* PyrKII

Since pyruvate does not seem to be required for apicoplast maintenance, we examined the nucleotide products of PyrKII. While PyrKII was previously localized to the apicoplast in *P. falciparum*, the enzyme has not been characterized (*Maeda et al., 2009*). The closely related apicomplexan *T. gondii* has a high substrate specificity for GDP, suggesting that GTP is the major product of this enzyme (*Saito et al., 2008*). Pyruvate kinase enzymes in other organisms have been shown to play a broader role in NTP generation and are potentially able to function as a general NDP kinase (*Saeki et al., 1974*). We hypothesized that if PyrKII has a low substrate specificity, it may be capable of generating several NTPs within the apicoplast.

We found that a codon harmonized version of PyrKII (*Table 1—source data 1*) could be expressed as a fusion protein with MBP (maltose-binding protein) in *E. coli*. We were able to cleave the MBP domain and obtain pure recombinant PyrKII for enzymatic characterization (*Table 1—source data 2*). We determined the kinetic parameters of PyrKII using a lactate dehydrogenase (LDH)-coupled spectrophotometric assay (*Kahn and Marie, 1982*). Our results show that PyrKII is fairly nonspecific, converting all the NDPs and dNDPs that were tested into the corresponding triphosphates (*Table 1*, *Table 1—source data 3* and *Table 1—source data 4*). In comparison, the $k_{cat}/K_m$ values for the *T. gondii* PyrKII were 337-fold higher for GDP than for ADP (*Saito et al., 2008*). However, for the *P. falciparum* PyrKII, the preference for GDP over ADP was less than two-fold, and the greatest difference in catalytic efficiency was between GDP and dCDP, which only differed by 38-fold. These results indicate that PyrKII lacks strong substrate specificity, and thus may play an important role in generating a variety of NTP and dNTP products in the apicoplast.

## Tracking loss of the apicoplast after mislocalization of PyrKII

While PyrKII is essential for apicoplast maintenance, and can generate NTPs and dNTPs, it is not clear if loss of PyrKII results in the inhibition of apicoplast NTP/dNTP production. Genetic deletion of PyrKII in the PfMev line resulted in apicoplast disruption, but these deletion lines do not allow us to observe what led to the loss of the organelle. If PyrKII is critical for NTP production within the apicoplast, deletion of PyrKII should result in a rapid reduction of NTPs that would precede apicoplast disruption. In order to study the sequence of events leading to organelle loss, a conditional localization domain (CLD) was appended to the N-terminus of the endogenous PyrKII (*Figure 5a and b*). The CLD directs trafficking of the protein to the apicoplast, but will result in secretion into the parasitophorous vacuole (PV) upon the addition of the ligand Shield1 (*Roberts et al., 2019*). Growth curve

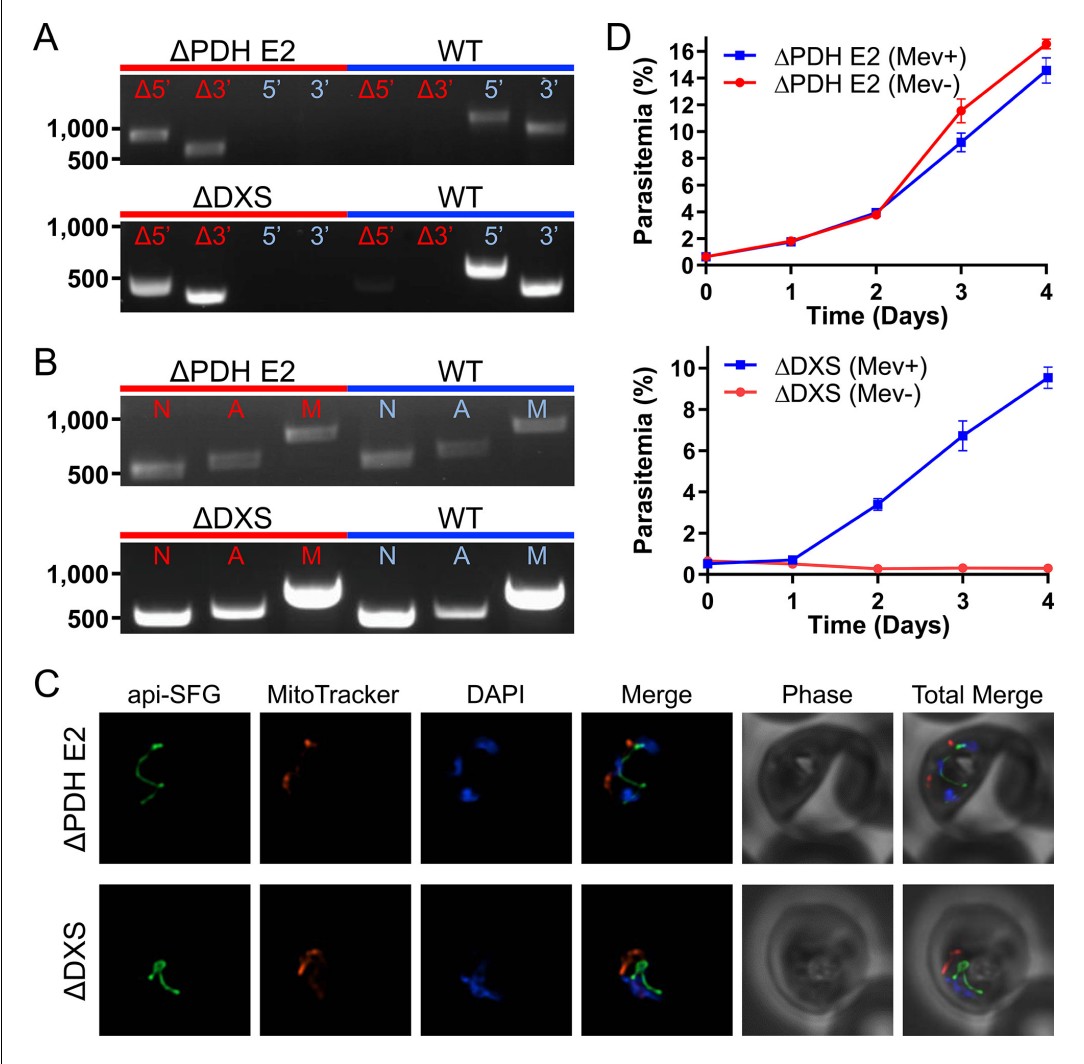

**Figure 4.** Characterization of the PfMev Δ*pdh E2* and Δ*dxs* parasite lines. (**A**) Genotyping PCR confirming deletion of PDH E2 or DXS, with amplification demonstrating drug cassette integration at the 5′ and 3′ loci of the target gene (Δ5′ and Δ3′), and lack of wild-type parasites due to failure to amplify at the wild-type 5′ and 3′ loci in the PfMev Δ*pdh E2* or Δ*dxs* parasites, as compared to the parental control (WT). (**B**) Attempted PCR detection of the *ldh*, *sufB*, and *cox1* genes from the nuclear (**N**), apicoplast (**A**), and mitochondrial (**M**) genomes, respectively. We successfully amplified *sufB* from the PfMev Δ*pdh E2* and Δ*dxs* parasite lines, indicating retention of the apicoplast organelle genome. (**C**) Live epifluorescence microscopy of the PfMev Δ*pdh E2* and Δ*dxs* parasite lines. These parasite lines express api-SFG (green) and are also stained with MitoTracker (red) and DAPI (blue). Microscopy images represent fields that are 10 µM long by 10 µM wide. (**D**) Growth curves of the PfMev Δ*pdh E2* and Δ*dxs* parasite lines cultured with or without 50 µM mevalonate. Error bars represent the standard error of the mean from two independent experiments, each conducted in quadruplicate.

determination of the PfMev CLD-PyrKII line demonstrated that the addition of Shield1 resulted in a reduction of parasite growth, beginning at 3 days post-induction (*Figure 5c*). We found that growth inhibition occurred faster using the CLD system, compared to the previously established TetR-DOZI knock down system (*Ganesan et al., 2016*), in which a reduction in growth could not be observed until 5 days post-induction (*Figure 5—figure supplement 1*). Thus, the PfMev CLD-PyrKII line was used for all subsequent experiments.

To determine the effect on the apicoplast, mislocalization of PyrKII was induced in synchronized ring stage parasites and the morphology of the apicoplast was assessed via epifluorescence microscopy every ~24 hr for 5 days. Apicoplast disruption occurred ~88–110 hr (4–5 days) post-mislocalization (*Figure 6a*). The percentage of parasites containing an intact organelle was ~90–100% until the 64 hr time point, and then dropped precipitously in the transition between the second and third 48 hr growth cycle. Approximately 25% of the parasites contained an intact apicoplast at the 88 hr (day

**Table 1.** Enzyme kinetics of pure recombinant PyrKII as compared to *T. gondii* PyrKII.
Activity of PyrKII was measured using various NDPs and dNDPs as substrates. Values and standard deviations from triplicate measurements are reported for the kinetic parameters $K_{cat}$ and $K_m$. The ratio $K_{cat}/K_m$ represents the catalytic efficiency. Kinetic parameters for *T. gondii* PyrKII from Saito and coworkers are shown for comparison (**Saito et al., 2008**).

| Substrate | $K_{cat}$ (min$^{-1}$) | $K_m$ (mM) | $K_{cat}/K_m$ (mM$^{-1}$ min$^{-1}$) | Organism |
|---|---|---|---|---|
| ADP | 793 + / - 186 | 0.25 + / - 0.11 | 3200 | *P. falciparum* |
| GDP | 397 + / - 45 | 0.07 + / - 0.04 | 5890 | |
| CDP | 438 + / - 69 | 1.01 + / - 0.13 | 432 | |
| UDP | 252 + / - 40 | 0.82 + / - 0.22 | 306 | |
| dADP | 401 + / - 31 | 0.38 + / - 0.04 | 1066 | |
| dGDP | 90 + / - 5 | 0.28 + / - 0.03 | 318 | |
| dCDP | 78+ / - 3 | 0.52 + / - 0.12 | 149 | |
| ADP | 2850 + / - 80 | 8.0 + / - 0.40 | 360 | *T. gondii* |
| GDP | 6600+ / - 240 | 0.05 + / - 0.006 | 121,320 | |

The online version of this article includes the following source data for Table 1:

Source data 1. Nucleotide sequence of *P. falciparum* PyrKII codon harmonized for expression in *E. coli*. The endo-nuclease sites used for cloning (EcoRI and HindIII) are shown in lower case italics and the stop codon is highlighted in red.

Source data 2. Purification of recombinant *P. falciparum* PyrKII. A maltose binding protein (MBP), along with a histidine (His) tag were appended to the N-terminus of harmonized PyrKII sequence and expressed in *E. coli*. The MBP-His-PyrKII fusion protein (126,273 Da) can be seen in lane 7, after purification using an MBP column. The TEV cleaved product, His-PyrKII (83,014 Da) can be seen in lane 8. Lane 10 shows the purified product and a minor band consistent with some covalent dimer (166 kDa).

Source data 3. Nucleotide substrate kinetic plots for His-PyrKII. Dose-response curves from triplicate data for seven nucleotide substrates are shown fitted with hyperbolic curves based on the Michaelis-Menton equation. Insets show the same data plotted in double reciprocal (Lineweaver-Burk) plots. Error bars represent the standard error of the mean and the program Prism (GraphPad) was used to generate the plots in this figure.

Source data 4. Nucleotide substrate kinetic plots for His-PyrKII. Dose-response curves from triplicate data for seven nucleotide substrates are shown fitted with hyperbolic curves based on the Michaelis-Menton equation. Insets show the same data plotted in double reciprocal (Lineweaver-Burk) plots. Error bars represent the standard error of the mean and the program Prism (GraphPad) was used to generate the plots in this figure.

4) time point, and less than 5% of the parasites contained an intact apicoplast at the 110 hr (day 5) time point (*Figure 6b*). Additionally, during this same experiment, RNA levels were measured as a proxy for NTP levels every 8 hr via microarray. The RNA levels of the Shield1-treated PfMev CLD-PyrKII parasites were plotted relative to control parasites. Apicoplast RNA levels drop approximately 24–40 hr after triggering mislocalization of PyrKII, whereas nuclear and mitochondrial RNA levels were not significantly reduced during this time period (*Figure 6c*). The most rapid decrease was observed with apicoplast mRNAs, perhaps a consequence of tRNAs and rRNAs forming relatively more stable folded structures (*Figure 6—figure supplement 1*).

The decrease in apicoplast RNA levels preceded organelle disruption, suggesting that inhibition of apicoplast gene transcription is an early consequence of interfering with PyrKII activity. To support this conclusion, we asked whether interfering with an unrelated apicoplast pathway would also perturb apicoplast gene transcription. Using real-time quantitative PCR (RT-qPCR), we tracked the levels of nuclear and apicoplast RNAs in the PfMev Δ*dxs* parasite line after loss of the mevalonate supplement required for survival. Transcription was not significantly altered over 48 hr, indicating that loss of isoprenoid synthesis does not mimic the effect we observed after triggering mislocalization of PyrKII (*Figure 6—figure supplement 2*).

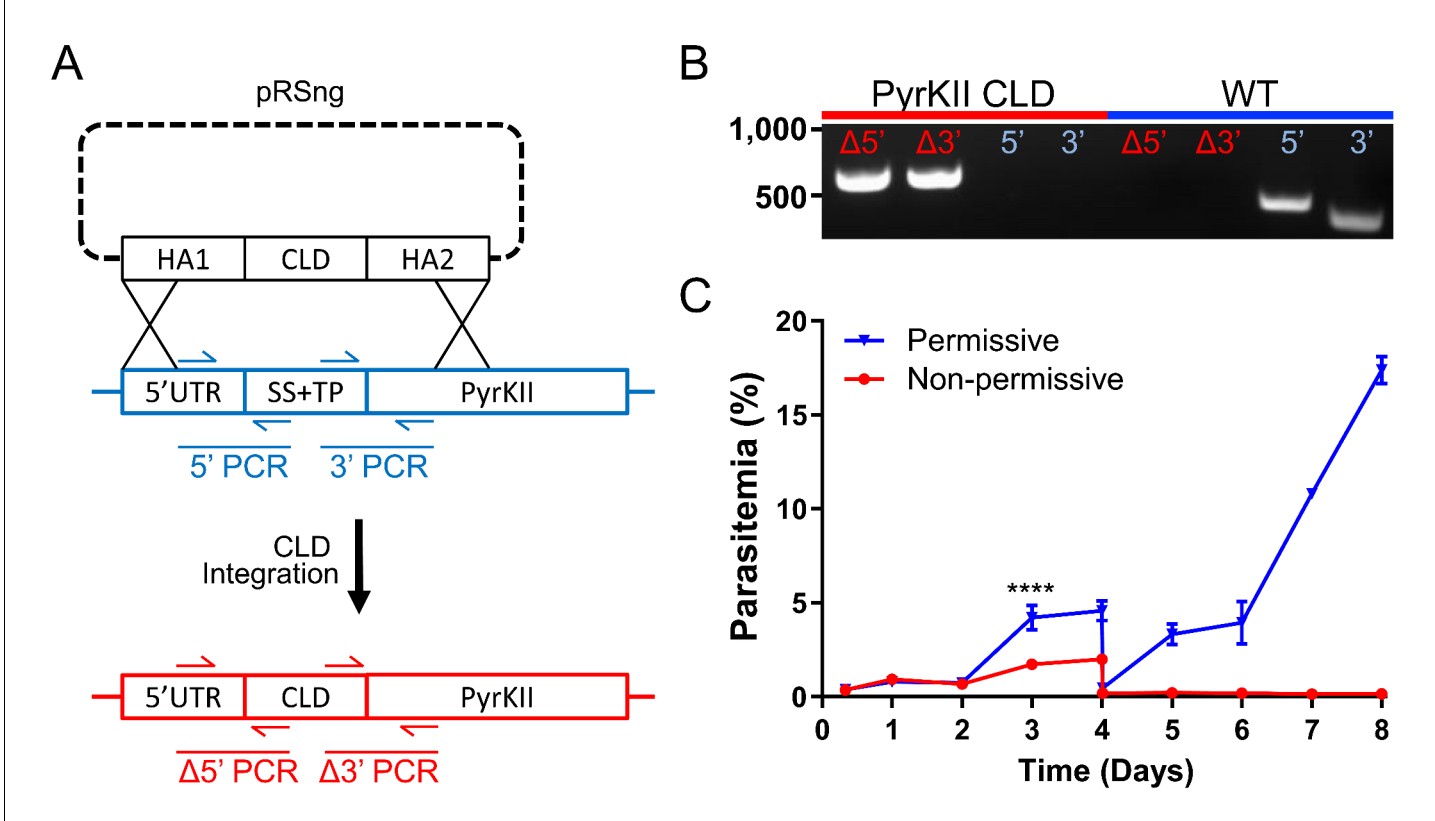

**Figure 5.** Growth of the PfMev CLD-PyrKII line under permissive and non-permissive conditional localization conditions. (**A**) Schematic representation showing how the endogenous signal sequence (SS) and transit peptide (TP) of PyrKII were replaced with the CLD using plasmid pRSng. The positions of genotyping PCR products designed to identify the wild type locus (blue) or the recombinant locus (red) are shown. (**B**) Amplification of the 5' and 3' loci of the integrated region (Δ5' and Δ3') demonstrates integration of the CLD (PyrKII CLD), and lack of wild type parasites due to failure to amplify at the wild type 5' and 3' regions of the endogenous gene, as compared to the parental control (WT). (**C**) Synchronized PfMev CLD-PyrKII parasites were seeded at 0.5% parasitemia and cultured in the permissive (no Shield1) or non-permissive (500 nM Shield1) conditions for 8 days. Parasites were collected daily for determination of parasitemia by flow cytometry and were cut 1:10 on day 4. Treatment of the parasites with Shield1 resulted in a relatively rapid inhibition of parasite growth beginning at day 3 (two-way ANOVA, followed by Bonferroni's correction; ****, p<0.0001). Error bars represent the standard error of the mean from two independent experiments, each conducted in quadruplicate.

The online version of this article includes the following figure supplement(s) for figure 5:

**Figure supplement 1.** Growth of the PfMev PyrKII TetR-DOZI line under permissive and non-permissive knockdown conditions.

## Tracking ATP levels in the apicoplast after mislocalization of PyrKII

In order to more directly demonstrate that PyrKII is important for NTP generation within the apicoplast, we conducted an additional experiment in which we directed the trafficking of firefly luciferase to the apicoplast in the PfMev CLD-PyrKII parasite line to measure ATP levels in the organelle upon the mislocalization of PyrKII. Trafficking of luciferase to the apicoplast was mediated by appending the N-terminal signal and transit peptide of ACP to the N-terminus of luciferase and correct localization was verified using a C-terminal mCherry tag (*Figure 6—figure supplement 3*). This luciferase construct was under the control of the TetR-DOZI knockdown system (*Ganesan et al., 2016*) and, as expected, removing aTc resulted in reduced levels of mCherry (*Figure 6—figure supplement 3*). Luciferase has been previously used as an ATP biosensor (*Prioli et al., 1985*), with changes in luminescence associated with changes in the level of ATP (*Morciano et al., 2017*). Once trafficking of luciferase to the apicoplast was confirmed, we synchronized the PfMev CLD-PyrKII api-luciferase parasite line and treated the culture with Shield1 to induce mislocalization of PyrKII from the apicoplast to the PV. We then analyzed the culture every 8 hr for 48 hr, measuring the change in luminescence over time (*Figure 6d*). We observed a significant reduction in luminescence approximately 8–24 hr after triggering mislocalization of PyrKII. As a control, we also treated the parasites with the

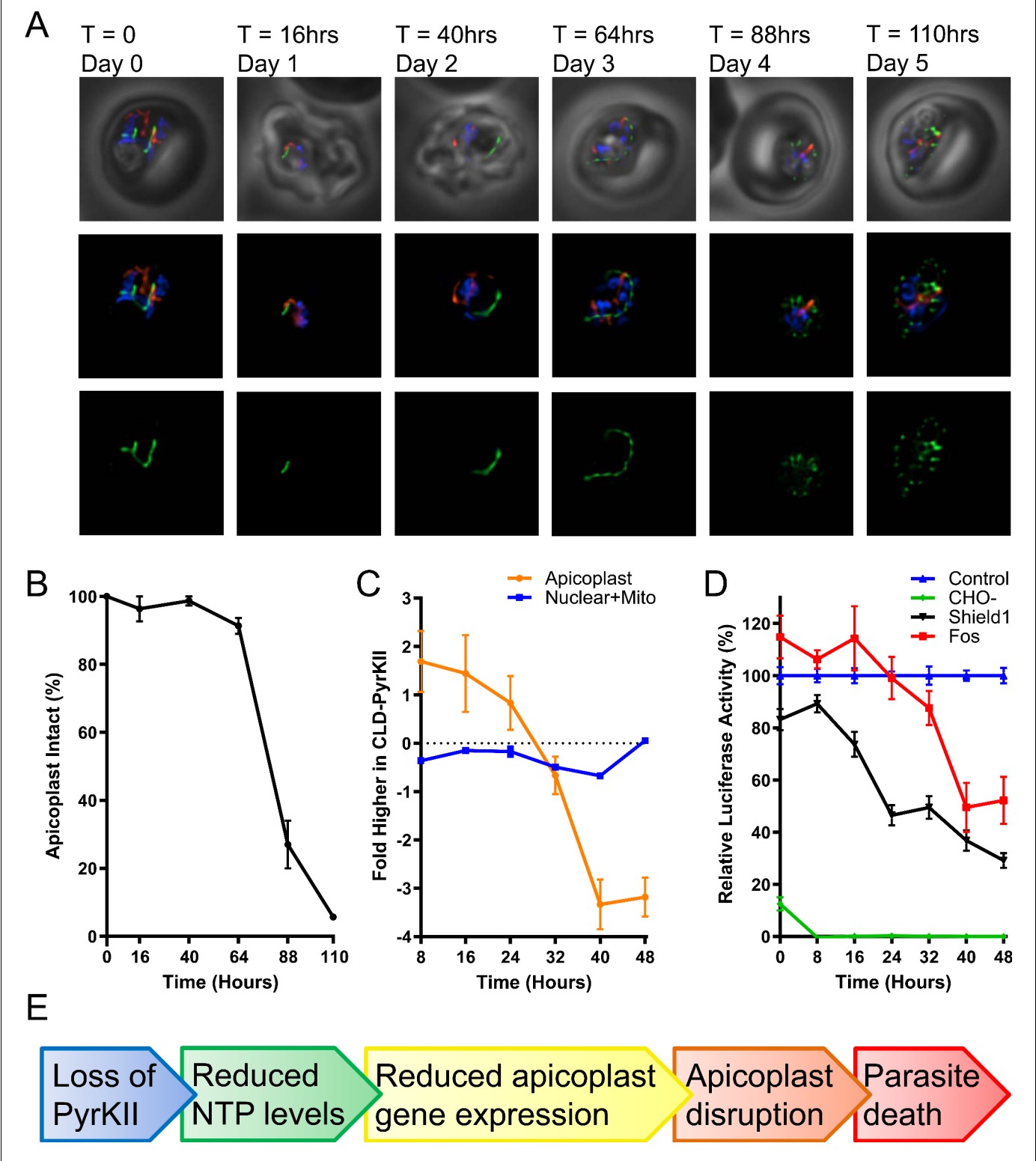

**Figure 6.** Characterization of PfMev CLD-PyrKII parasites after mislocalization of PyrKII. (**A**) Live epifluorescence microscopy of the PfMev CLD-PyrKII parasite line after Shield1 treatment. This parasite line expresses api-SFG (green) and is also stained with MitoTracker (red) and DAPI (blue). Representative images show the state of the apicoplast organelle at the corresponding time points. Microscopy images represent fields that are 10 μM long by 10 μM wide. (**B**) Graph of the percentage of disrupted apicoplast organelles in synchronized PfMev CLD-PyrKII parasites after Shield1 treatment. At each time point, a minimum of 15 microscopy images were taken to determine apicoplast morphology (intact or disrupted) based on api-

*Figure 6 continued on next page*

*Figure 6 continued*

SFG fluorescence. The percentage of parasites containing an intact apicoplast dropped precipitously between the second and third growth cycle of the parasite, corresponding to the 64 and 88 hr time points. Error bars represent the standard deviation from three independent experiments with at least 20 observations each. (C) Synchronized ring-stage PfMev CLD-PyrKII parasites were treated with Shield1, with samples collected every 8 hr for microarray analysis (biological triplicates were collected). Nuclear and mitochondrial mRNA levels (blue) were compared to an untreated control and plotted as the fold difference. The same analysis for apicoplast genes (orange) shows that there is a sharp reduction in RNA levels in the treated parasites between 24 and 40 hr after Shield1 treatment. (D) Synchronized ring-stage PfMev CLD-PyrKII api-luciferase parasites were prepared for a 48 hr time course under four conditions, each of which contained 500 nM aTc: exchange into glucose-free medium (CHO-, green), 500 nM Shield1 treatment (Shield1, black), 50 μM fosmidomycin treatment (Fos, red), and no treatment (Control, blue). Samples were collected every 8 hr for analysis of luciferase activity (biological triplicates were collected, each with three technical replicates). We observed a significant reduction in luminescence ~8–24 hr after triggering mislocalization of PyrKII with Shield1. For parasites treated with fosmidomycin, luminescence levels did not meaningfully decrease until 24–40 hr, while levels in parasites cultured in glucose-free medium dropped within 8 hr. For all treatment groups, luminescence levels were compared to an untreated parasite control. Error bars represent the standard error of the mean. (E) Proposed progression of events leading from loss of PyrKII to parasite death.

The online version of this article includes the following figure supplement(s) for figure 6:

**Figure supplement 1.** Analysis of apicoplast RNA levels in PfMev CLD-PyrKII parasites after treatment with Shield1.

**Figure supplement 2.** Mevalonate dependence of apicoplast and nuclear transcripts in PfMev Δ*dxs* parasites quantified using RT-qPCR.

**Figure supplement 3.** Design and validation of the PfMev CLD-PyrKII api-luciferase parasite line.

**Figure supplement 4.** Sequences found in plasmid p15-p230-aFluc-mCh.

apicoplast inhibitor fosmidomycin (Fos) at 100x its $IC_{50}$ value (without addition of mevalonate) and observed a reduction in luminescence at later time points (24–40 hr). Transferring the parasites into glucose-free medium (CHO-) resulted in a rapid loss of luminescence, consistent with apicoplast ATP production being dependent on the glycolytic product PEP (*Figure 1*). Taken together, these results suggest that PyrKII is involved in the production of NTPs, which are required for apicoplast gene transcription and organelle maintenance (*Figure 6e*).

## Discussion

The work outlined here has helped to better elucidate the role and importance of carbon metabolism within the apicoplast organelle. This work began by demonstrating that carbon import into the apicoplast, mediated by the outer and inner triose phosphate transporters (oTPT/iTPT) (*Lim et al., 2010*; *Ralph et al., 2004*), is required for apicoplast maintenance and parasite survival. Our findings for oTPT are consistent with previous work demonstrating that oTPT is required for blood-stage *P. berghei* survival (*Banerjee et al., 2012*), with a separate forward genetic screen suggesting that it is also essential in blood-stage *P. falciparum* (*Zhang et al., 2018*). However, our results for the deletion of iTPT are at odds with previous results showing that iTPT is dispensable in blood-stage *P. berghei* parasites (*Banerjee et al., 2012*). This inconsistency may be due to differences in metabolic requirements, or it could be due to the relative roles that oTPT and iTPT play in blood-stage *P. berghei* and *P. falciparum* parasites. However, the results found in this work are consistent with forward genetic screens conducted in *P. berghei* (*Schwach et al., 2015*) and *P. falciparum*, which suggest that iTPT is actually likely to be essential (*Zhang et al., 2018*).

Once PEP and DHAP are imported into the apicoplast, they are used as substrates by PyrKII and TPI, respectively. TPI is believed to convert DHAP into GA3P, providing the MEP pathway with one of its two initial substrates (*Ralph et al., 2004*). While the MEP pathway is essential, we demonstrated that TPI is dispensable. The dispensability of TPI can potentially be attributed to the lack of specificity of the upstream oTPT/iTPT transporters. Homologous transporters in plant plastids are relatively nonspecific and can import various triose phosphate compounds (*Fischer et al., 1997*). The *P. falciparum* oTPT and iTPT triose phosphate transporters were also demonstrated to be relatively nonspecific when tested in vitro, capable of importing 3-phosphoglyceric acid (3PGA), a compound not predicted to be used within the organelle (*Lim et al., 2010*). Thus, it is possible that the oTPT/iTPT transporters can import GA3P directly, bypassing the need for the conversion of DHAP to GA3P by TPI (*Figure 7*).

Previously, TPI was also thought to be involved in the generation of reducing power within the apicoplast working along with a hypothetical GAPDH in a two-enzyme cycle. In this cycle, TPI would convert DHAP into GA3P, which would be used by GAPDH to generate 1,3-diphosphoglycerate

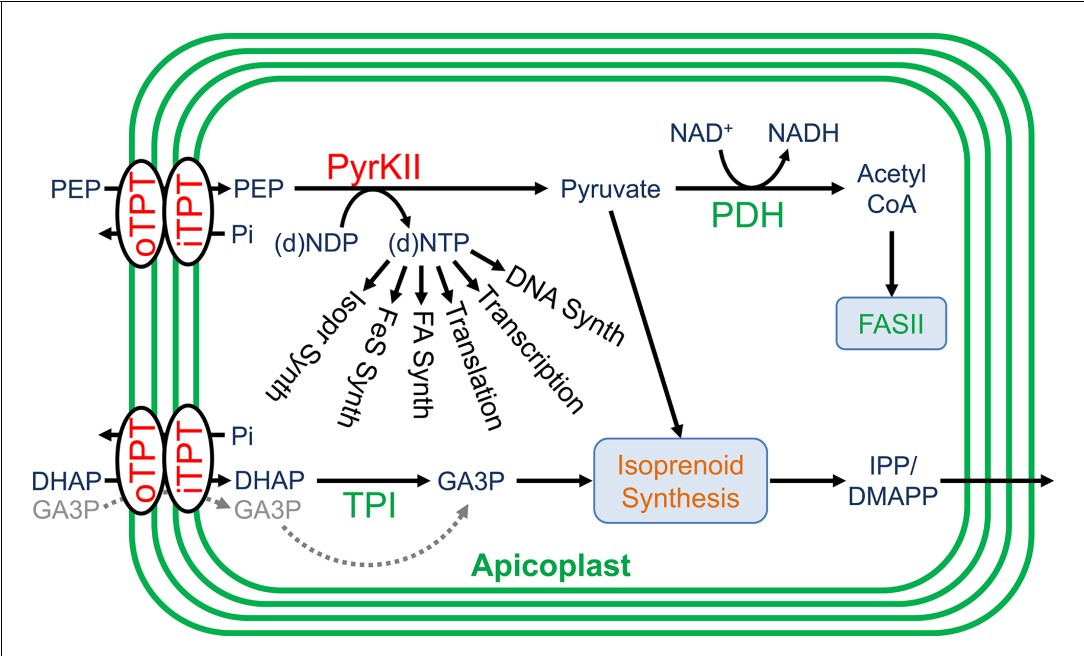

**Figure 7.** Roles of carbon metabolism proteins in the apicoplast. Apicoplast proteins and pathways are colored according to gene deletion results (dispensable = green, essential = orange, and essential for apicoplast maintenance = red). Pyruvate kinase II (PyrKII) generates nucleoside triphosphate (NTP) products used for transcription, translation, and for several metabolic pathways including the synthesis of fatty acids (FA), isoprenoids (isopr) and iron-sulfur clusters (FeS). The dispensability of the triose phosphate isomerase (TPI) suggests that glyceraldehyde-3-phosphate (GA3P) may be directly imported by oTPT/iTPT for use in the isoprenoid synthesis pathway (gray arrow). This result also indicates that malaria parasites do not rely on a two-enzyme reductive pathway involving TPI and a hypothetical GAPDH enzyme.

(1,3-DPGA), converting $NAD^+$ to NADH in the process (*Fleige et al., 2007*; *Waller et al., 1998*). The 1,3-DPGA would then presumably be exchanged for DHAP by oTPT/iTPT, forming an electron shuttle (*Waller et al., 1998*). This putative cycle would provide reducing power in the form of NADH or NADPH within the organelle, with the only other predicted sources being PDH and glutamate dehydrogenase (*Ralph et al., 2004*; *Zocher et al., 2012*; *Laine et al., 2015*). We were able to demonstrate that TPI is dispensable in blood-stage parasites. Additionally, the only GAPDH in *P. falciparum* has been shown to localize to the cytosol of the parasite (*Daubenberger et al., 2003*). Thus, this cycle is not essential and is likely not even present within the apicoplast.

PyrKII is the only predicted source of pyruvate within the apicoplast (*Lim et al., 2010*). We expected PyrKII to be essential for the generation of pyruvate needed by the MEP pathway, but we were surprised to find that PyrKII is required for apicoplast maintenance. To understand this phenomenon, we deleted the two proteins predicted to use pyruvate in the apicoplast, DXS and PDH. Deletion of DXS from the MEP pathway demonstrated that this protein is essential for parasite survival but is not required for organelle maintenance. Deletion of the PDH E2 subunit, which converts pyruvate into acetyl-CoA (*Foth et al., 2005*), revealed that this gene is not required for blood-stage parasite survival or apicoplast maintenance. The dispensability of PDH replicates what has been seen previously with the deletion of the PDH E1α and E3 subunits in *P. yoelli* (*Pei et al., 2010*) and the PDH E1α (*Cobbold et al., 2013*) and E3 (*Laine et al., 2015*) subunits of *P. falciparum*. The dispensability of both TPI and PDH is interesting because these are two of the few predicted sources of reducing power within the organelle (*Ralph et al., 2004*; *Zocher et al., 2012*; *Foth et al., 2005*). The dispensability of these proteins indicates that other sources of reducing power are capable of compensating for their loss during this stage of the parasite cycle. Overall, these results suggest that the requirement of PyrKII for organelle maintenance is not driven by a need for pyruvate by downstream metabolic pathways.

The findings for PyrKII in *P. falciparum* do not hold for all apicomplexan species, as the apicoplast-localized PyrKII has been shown to be dispensable in *T. gondii* parasites without any observed growth phenotype (*Xia et al., 2019*). Pyruvate should be required for fatty acid (FASII) and

isoprenoid precursor synthesis in the apicoplast of *T. gondii*. The FASII pathway was thought to be essential for apicoplast maintenance and parasite survival (*Mazumdar et al., 2006*; however, recent studies show that several PDH subunits and FASII enzymes can be deleted; *Krishnan et al., 2020*; *Liang et al., 2020*). These FASII-null parasite lines grow poorly in vitro and one would expect the PyrKII deletion line to share this phenotype unless the apicoplast has some other source of pyruvate. Similarly, work on isoprenoid synthesis enzymes suggests that a source of apicoplast pyruvate is essential for parasite survival. Although *T. gondii* parasites can scavenge some host cell isoprenoids (*Blume and Seeber, 2018*; *Imlay and Odom, 2014*; *Li et al., 2013*), the apicoplast-localized synthesis enzyme LytB was shown to be essential (*Nair et al., 2011*). Thus, both the FASII and isoprenoid precursor synthesis pathways are required for normal in vitro growth of *T. gondii* tachyzoites, whereas PyrKII is not. As suggested previously (*Xia et al., 2019*), the dispensability of PyrKII may indicate that pyruvate can be imported into the apicoplast of *T. gondii* parasites.

Our work with PyrKII led us to focus on its role in nucleotide metabolism in the apicoplast. To investigate which NTPs or dNTPs PyrKII may be capable of generating, we determined the substrate specificity of pure recombinant protein. These assays demonstrated that PyrKII has a relatively broad substrate specificity, and the $K_m$ values shown in *Table 1* could be consistent with the levels of NDPs in the apicoplast. The concentrations of nucleotides in the apicoplast have not been determined; however, NDP concentrations of 0.1–1 mM are typical for other cells and tissues (*Traut, 1994*). Thus, PyrKII may be capable of generating a variety of nucleoside triphosphates within the apicoplast. This finding is not unprecedented, as other pyruvate kinase enzymes have been shown to be relatively non-specific in their use of nucleotide substrates (*Saeki et al., 1974*; *Abbe and Yamada, 1982*; *Boehme et al., 2013*; *Knowles et al., 2001*; *Plowman and Krall, 1965*). We hypothesized that if PyrKII is required for NTP generation, then the loss of this enzyme would result in a decrease in mRNA levels in the apicoplast. To probe this question, we created two different parasite lines that allow for conditional control of the endogenous PyrKII enzyme. Parasites incorporating the aptamer-based knockdown method of Ganesan and coworkers (*Ganesan et al., 2016*) displayed a growth defect 5 days after triggering the knockdown of PyrKII (*Figure 5—figure supplement 1*), whereas parasites incorporating a conditional localization (CLD) approach (*Roberts et al., 2019*) grew poorly 3 days after initiating mislocalization (*Figure 5*). We used the CLD-PyrKII parasites to quantify mRNA levels during the 2-day period following PyrKII mislocalization. We found that apicoplast transcript levels dropped ~24–40 hr after PyrKII mislocalization, whereas nuclear transcript levels were not affected during this time period (*Figure 6c*). Importantly, this drop in apicoplast transcript levels preceded disruption of apicoplast morphology by over 50 hr, which was not observed until ~88–110 hr after mislocalization. This demonstrated that reduced apicoplast gene expression is an early event in the process of apicoplast disruption. This is in contrast to results from the analysis of apicoplast RNA levels in PfMev Δ*dxs* parasites, which did not drop significantly upon removal of mevalonate and loss of isoprenoid synthesis (*Figure 6—figure supplement 2*).

Experiments with firefly luciferase strengthened the hypothesis that PyrKII generates NTPs in the apicoplast. We expressed luciferase in the apicoplast of PfMev CLD-PyrKII parasites and used it to probe ATP levels. Cellular ATP levels are known to fall rapidly in malaria parasites deprived of glucose (*Teng et al., 2009*). We observed significantly decreased luminescence after a short period in glucose-free medium (~20 min), linking parasite glycolysis to the activity of apicoplast luciferase (*Figure 6d*). Presumably, the glycolysis product PEP is required for the production of ATP by PyrKII in the apicoplast (*Figure 7*). Treatment with the isoprenoid biosynthesis inhibitor fosmidomycin has been shown to kill parasites within one 48 hr life cycle (*Jomaa et al., 1999*). We observed a reduction in luminescence 24–40 hr after treatment with 100x $IC_{50}$ of fosmidomycin, consistent with parasite death during the second half of the parasite life cycle. Importantly, mislocalization of PyrKII resulted in lowered luminescence during the first half of the life cycle (8–24 hr), demonstrating that lowered ATP is an early consequence of reduced PyrKII activity. It is not clear why luciferase activity didn't drop to zero after PyrKII mislocalization or fosmidomycin-treatment. It may be that other essential nutrients are depleted before apicoplast ATP is fully consumed.

Inhibition of NTP production within the apicoplast could result in parasite death through various molecular mechanisms due to the requirement of NTPs for a number of biochemical processes within the organelle. For example, within the MEP pathway the enzyme IspD uses CTP along with 2-C-methyl-D-erythritol 4-phosphate (MEP) to generate 4-diphosphocytidyl-2-C-methyl-D-erythritol (CDP-ME), which goes on to form the essential isoprenoid precursors IPP and DMAPP

(*Hunter, 2007*). GTP is required for multiple steps of protein translation including translation initiation, peptide chain elongation, translation termination, and ribosome recycling, which are processes mediated by translational GTPases (*Maracci and Rodnina, 2016*). ATP is required for tRNA charging by aminoacyl-tRNA synthetases (*Pang et al., 2014*), iron-sulfur cluster formation by the SufC ATPase (*Saini et al., 2010*), DNA gyrase subunits (*Mukherjee and Sadhukhan, 2016*), as well as numerous other energy requiring pathways within the organelle (*Lim and McFadden, 2010*). Additionally, NTPs and dNTPs are required for transcription and DNA replication, respectively (*Joyce and Steitz, 1995*; *Nudler, 2009*). Thus, inhibiting the generation of these products would ultimately cause several important biochemical processes to fail within the organelle (*Figure 7*).

For plastid organelles ATP/ADP transport is typically mediated by a nucleotide transporter (NTT) (*Fischer, 2011*). While NTT is widespread in plastid organelles this transporter has not been found in *Plasmodium* (*Facchinelli and Weber, 2011*). With no other sources of ATP identified, this has led to the hypothesis that PyrKII is responsible for generating all ATP required within the organelle (*Lim and McFadden, 2010*).

While the NTP generating activity of PyrKII appears to be required for organelle maintenance and parasite survival, the import mechanism and the phosphorylation state (NMP, NDP, or NTP) of nucleotides remain unknown. We found that PyrKII is active against NDPs, but it has essentially no activity against NMPs. This suggests that nucleotides need to be imported as NDPs, or additional mechanisms for converting NMPs into NDPs would need to be present within the apicoplast. A putative UMP-CMP kinase (PF3D7_0111500) with predicted apicoplast localization could convert at least some NMPs to NDPs, but its substrate specificity has not been determined (*Boucher et al., 2018*; *Zuegge et al., 2001*). None of the other nucleotide kinases, purine interconversion and pyrimidine biosynthetic enzymes are predicted to localize to the apicoplast. Additionally, only single copies of the ribonucleotide reductase subunit genes and thymidylate synthase are present in *Plasmodium*. These enzymes presumably function in the cytoplasm to supply dNDPs for nuclear and organellar DNA synthesis. Thus, nucleotide transport should also be required for these deoxyribose nucleotides. Overall, nucleotide import and metabolism in the apicoplast are poorly studied and remain to be fully elucidated.

Taken together, the results herein help elucidate the role and importance of carbon metabolism within the apicoplast of *P. falciparum* parasites. We showed that PDH and TPI are dispensable for asexual parasite survival, raising additional questions as to how reducing power is sufficiently generated in the absence of these enzymes. Importantly, this work also demonstrated that PyrKII is essential for parasite survival and apicoplast maintenance. In addition to PyrKII being essential for *P. falciparum* parasite survival, PyrKII enzymes are also evolutionary distinct, and appear to occur only within apicomplexans, and thus may represent an attractive drug target for *Plasmodium* parasites (*Chan et al., 2007*). While the pyruvate generated by PyrKII is required to generate essential isoprenoid precursors in the MEP pathway, our results indicate that pyruvate is not needed for apicoplast maintenance. Instead, it appears that the requirement of PyrKII for organelle maintenance is driven by its role in NTP production within the apicoplast, with loss of this activity likely being the cause of subsequent apicoplast disruption. Loss of PyrKII activity in the apicoplast results in a reduction of apicoplast mRNA, tRNA and rRNA significantly prior to organelle disruption and parasite death. These observations provide new insight into carbon metabolism and the generation of reducing power within the apicoplast and demonstrate an unexpected role for PyrKII in apicoplast maintenance.

## Materials and methods

### *P. falciparum* culture and maintenance

Unless otherwise noted, blood-stage *P. falciparum* parasites were cultured in human erythrocytes at 1% hematocrit in a 10 mL volume of CMA (Complete Medium with Albumax) containing RPMI 1640 medium with L-glutamine (USBiological Life Sciences), supplemented with 20 mM HEPES, 0.2% sodium bicarbonate, 12.5 µg/mL hypoxanthine, 5 g/L Albumax II (Life Technologies) and 25 µg/mL gentamicin. Cultures were maintained in 25 cm$^2$ gassed flasks (94% $N_2$, 3% $O_2$, 3% $CO_2$) and incubated at 37°C. The PfMev parasite line (*Swift et al., 2020*) was used extensively in this project. This line expresses an apicoplast-localized green fluorescent protein and four enzymes capable of

producing isoprenoid precursors from mevalonate, bypassing the need for the endogenous MEP isoprenoid pathway.

## Generation of *P. falciparum* plasmid constructs for gene deletion

Target genes were deleted through Cas9-mediated gene editing methods developed by Dr. Jose-Juan Lopez-Rubio (*Ghorbal et al., 2014*). Homology repair plasmids pL8 or pRS (*Swift et al., 2020*) were used in combination with the Cas9-expressing pUF1-Cas9 plasmid (*Ghorbal et al., 2014*). Alternatively, genes were targeted for deletion using the pRSng plasmid in combination with the pCasG plasmid (*Swift et al., 2020*; *Rajaram et al., 2020*). The pRSng plasmid is a modified version of the pRS plasmid in which the guide RNA expression cassette was removed, and is thus reliant on gRNA expression from the pCasG plasmid. For generation of the deletion constructs, homology arms of ~300–600 bp were amplified from *P. falciparum* NF54 strain gDNA using the homology arm (HA) 1 and 2 forward and reverse primers corresponding to each gene (*Supplementary file 1*), and inserted into the repair plasmids using ligation-independent cloning (LIC) methods (In-Fusion, Clontech). The repair plasmids were digested with NotI for insertion of HA1, and with NgoMIV for insertion of HA2. Guide RNA sequences were synthesized as oligos (*Supplementary file 1*), annealed, and inserted using In-Fusion.

## *P. falciparum* transfections for gene deletion

Transfections were conducted as previously described (*Spalding et al., 2010*). Briefly, 400 µL of red blood cells (RBCs) were electroporated with 75 µg each of the Cas9 expression plasmid and the corresponding homology repair plasmid. The transfected RBCs were mixed with synchronized schizont stage PfMev parasites and maintained in 10 mL CMA containing 50 µM mevalonate. After ~48 hr, 1.5 µM DSM1 and 2.5 nM WR99210 were added to ensure the retention of both transfection plasmids, in the presence of 50 µM mevalonate. After 7 days of drug selection, the parasites were switched to media containing only 50 µM mevalonate. Infected red blood cells (iRBC) were first observed 17 to 25 days after beginning drug selection. Once parasites were observed, 2.5 nM WR99210 was reintroduced into the growth medium to maintain the integrated hDHFR drug resistance cassette. Deletion of oTPT did not work with our first choice of gRNA, but was successful with a second gRNA sequence (*Supplementary file 1*). For each gene deletion, parasites from at least two independent transfections were characterized as described below.

## Confirmation of knockout genotype

Primers were designed to screen for 5′ integration (Δ5′ reaction primers GOI.5′.F and pL8HA1.R) and 3′ integration (Δ3′ reaction primers pL8HA2.F and GOI.3′.R) of the gene disruption cassette, and the 5′ region (primers GOI.5′.F and GOI.5′WT.R) and 3′ region (primers GOI.3′WT.F and GOI.3′.R) of the gene of interest (GOI) (*Supplementary file 1*). The parental PfMev line was used as a control for these reactions. Details about the genotyping PCR reactions, including the anticipated amplicon sizes, are provided in *Figure 2—figure supplement 1*.

## Confirmation of apicoplast loss

The presence of the apicoplast organellar genome was detected by PCR with primers specific for the *sufB* gene (PF3D7_API04700). Control PCR reactions amplified genes from the nuclear (lactate dehydrogenase, *ldh,* PF3D7_1324900) and mitochondrial (cytochrome c oxidase subunit 1, *cox1,* mal_mito_2) genomes (*Supplementary file 1*). For the PCR reactions, 1 µL of parasite culture was added to a 50 µL reaction volume. The parental PfMev line was used as a positive control for apicoplast genome detection. The expected amplicon sizes are 520 bp, 581 bp and 761 bp for *ldh*, *sufB* and *cox1*, respectively.

## Live cell epifluorescence microscopy

Approximately 100 µL of a resuspended parasite culture was incubated with 30 nM MitoTracker CMX-Ros (Invitrogen) and 1 µg/mL 4′, 6-diamidino-2-phenylindole (DAPI) for 30 min at 37°C. Cells were then washed three times with 100 µL of CMA media and incubated for 5 min at 37°C after each wash. Cells were resuspended in 20 µL of CMA and then pipetted onto slides and sealed with wax for observation on a Zeiss AxioImager M2 microscope. A series of images spanning 5 µm were

acquired with 0.2 μm spacing and images were deconvolved with VOLOCITY software (PerkinElmer) to report a single image in the z-plane.

## Testing mevalonate dependence in PfMev deletion mutants via flow cytometry

Gene deletions generated in the PfMev line were tested to determine reliance on mevalonate for survival. Parasite lines were washed with 10 mL CMA three times to remove any mevalonate from the culture medium and then cultured in CMA alone or CMA with 50 μM mevalonate. Parasites were seeded in a 96-well plate (Corning) at a 0.5% starting parasitemia and 2% hematocrit in a total volume of 250 μL, in quadruplicate for each condition. Plates were incubated in chambers gassed with 94% $N_2$, 3% $O_2$, 3% $CO_2$ at 37°C. Parasite samples were collected and the culture medium was exchanged daily for 4 days.

For growth curve determination, parasites were stained with SYBR Green (Invitrogen), and analyzed via flow cytometry. Parasitemia was counted on the same day as seeding, after which parasites were collected every 24 hr. Samples collected on days 1–3 were diluted 1:10 in PBS and stored in a 96-well plate at 4°C. On day 4, parasites were stained with SYBR Green by transferring 1 μL of parasite culture, or 10 μL of the 1:10 dilutions, to a 96-well plate containing 100 μL of 1x SYBR Green (Invitrogen) per well in PBS. Plates were incubated at room temperature for 30 min while shaking, under protection from light. Post-incubation, 150 μL of PBS was added to each well to dilute unbound SYBR Green dye. Samples were analyzed with an Attune Nxt Flow Cytometer (Thermo Fisher Scientific), with a 50 μL acquisition volume, and a running speed of 25 μL/minute with 10,000 total events collected.

## Generation of the PyrKII *E. coli* expression vector

The *P. falciparum* Pyruvate Kinase II (PF3D7_1037100) was codon harmonized for expression in *E. coli* (*Table 1—source data 1*). The gene was synthesized encoding amino acids 42–745 of the protein, corresponding to the predicted mature protein after cleavage of the apicoplast trafficking peptide. The codon harmonized sequence was amplified using the primers listed in *Supplementary file 1* and inserted into the EcoRI and HindIII sites of the pMAL-cHT expression vector (*Muench et al., 2003*) using LIC (In-Fusion, Clontech). The pMAL-cHT expression vector generates a fusion protein tagging the PyrKII protein with an N-terminal MBP (maltose binding protein) followed by a TEV (Tobacco Etch Virus) protease site and a six-histidine tag. Cleavage with TEV protease liberates a histidine-tagged PyrKII (*Muench et al., 2003*).

## Expression and purification of recombinant PyrKII from *E. coli*

The pMAL-cHT PyrKII protein was purified from BL21 Star (DE3) *E. coli* cells (ThermoFisher) also harboring the pRIL plasmid from BL21-CodonPlus-RIL cells (Agilent). Cells harboring both plasmids were grown in LB medium, and shaken at 220 rpm at 37°C until an $OD_{600}$ of 0.6 was reached. At this point, protein expression was induced with 0.4 mM isopropyl β-thiogalactoside (IPTG) and the cells were then incubated for an additional 4 hr at 27°C. Cells were lysed with 20 mM Tris-HCl at pH 7.5, 100 mM KCl, 1 mg/mL lysozyme (Sigma), 2.5 μg/mL DNAse I (Sigma), and 1 mM PMSF, followed by sonication. The MBP-PyrKII protein was purified from the resulting lysate using a 5 mL MBPTrap HP column (GE Healthcare), and eluted with 100 mM maltose. The resulting protein was then cleaved with TEV protease and further purified using a 5 mL HisTrap column (GE Healthcare). The protein was then eluted with an imidazole gradient, and peak fractions were collected, pooled, and concentrated for storage in 20 mM Tris-HCl (pH 7.5), 100 mM KCl, 10% glycerol at −80°C.

## PyrKII enzyme assays

The activity of the recombinant PyrKII protein was determined by a continuous assay coupled to lactate dehydrogenase (LDH) (*Saito et al., 2008*), monitoring the oxidation of NADH via spectrophotometer at 340 nm. The reaction mixture for assaying PyrKII activity contained 1 μL of the PyrKII enzyme, 5 mM PEP (neutralized to pH 7), 10 mM $MgCl_2$, 0.2 mM NADH, 2 units of rabbit muscle lactate dehydrogenase (Sigma), 26.5 mM Tris-HCl at pH 7.5, and 50 mM KCl. The PyrKII enzyme was diluted appropriately to ensure the reaction remained in the linear range. All reactions were initiated through the addition of a given NDP or dNDP. The concentrations assayed for ADP and GDP ranged

from 0 mM to 3.0 mM, specifically at 0 µM, 4 µM, 12 µM, 37 µM, 111 µM, 333 µM, 1 mM, and 3 mM. CDP, UDP, TDP, dATP, dGDP, dCTP, and dTDP were assayed at concentrations between 0 mM to 6 mM, specifically at 0 µM, 125 µM, 250 µM, 500 µM, 1 mM, 2 mM, 4 mM, and 6 mM. The $K_m$ for PEP (0.64+ / - 0.05 mM) was measured with 3 mM ADP and PEP concentrations ranging from 0 mM to 20 mM. All reactions were carried out in a final volume of 100 µL in triplicate in a 96-well plate and monitored for 20 min. The initial enzyme velocity was calculated from a tangent that was fitted to the reaction curve. Kinetic parameters were calculated from Lineweaver-Burk plots (*Table 1—source data 3* and *Table 1—source data 4*) and are presented as mean values with standard deviations.

## Generation of the tetR-DOZI PyrKII parasite line

The PyrKII tetR-DOZI line was generated as previously described (*Ganesan et al., 2016*), using the pMG74 plasmid, generously provided by Dr. Jacquin Niles (Massachusetts Institute of Technology). The PyrKII pMG74 plasmid was generated by using the primers in *Supplementary file 1*. Briefly, the PyrKtetR.HA1.F and PyrKtetR.HA1.R primers were used to amplify the region of the gene corresponding to homology arm 1 (HA1), and the PyrKtetR.HA2.F and PyrKtetR.HA2.R primers were used to amplify the region of the gene corresponding to homology arm 2 (HA2). These products were joined in a second PCR reaction using the PyrKtetR.HA2.F and PyrKtetR.HA1.R primers, creating a combined HA1-HA2 fragment separated by EcoRV sites. This fragment was inserted into the pMG74 plasmid at the AscI and AatII restriction sites using LIC methods (In-Fusion, Clontech). This plasmid was then linearized with EcoRV for transfection into PfMev parasites. A gene specific pCasG plasmid was also generated, through digestion with BsaI and insertion of the guide RNA. Guide RNAs were synthesized as oligos (*Supplementary file 1*), annealed, and inserted in the pCasG plasmid using In-Fusion.

The linearized PyrKII pMG74 plasmid along with the corresponding pCasG plasmid were co-transfected into the PfMev line, using the transfection protocol described above. The parasites were selected for 7 days with 2.5 µg/mL blasticidin, 1.5 µM DSM1, and 500 nM anhydrotetracycline (aTc). After 7 days, the drug pressure was removed and the parasites were continuously cultured in CMA media containing 500 nM aTc. Parasites were observed between 20 and 30 days post-transfection, at which point 2.5 µg/mL blasticidin was re-introduced. Resulting parasites were single-cloned through limiting dilution.

## Generation of the PyrKII conditional localization line

A conditional localization domain was appended to the endogenous PyrKII enzyme using markerless Cas9-mediated genome editing (*Ghorbal et al., 2014*). For insertion, a repair construct was generated with HA1 composed of the 250 nucleotides immediately upstream of the PyrKII gene, followed by the conditional localization domain (*Roberts et al., 2019*), and ending with HA2 composed of PyrKII nucleotides 124–373. This repair construct was designed to replace the first 41 amino acids of PyrKII (containing the signal peptide and most of the apicoplast transit peptide) with the CLD. A synthetic construct including HA1 and the CLD was generated with BsaI cloning sites for insertion of HA2. Primers PyrKCLD.HA2.F and PyrKCLD.HA2.R were used to amplify HA2 from gDNA and insert it into the BsaI sites using LIC. This repair construct was then amplified with the primers listed in *Supplementary file 1* (PyrK.pRS.HA1.F, and PyrK.pRS.HA2.R) and inserted into a NotI cut pRSng plasmid using LIC (In-Fusion, Clontech). A gene-specific pCasG plasmid was generated as well. Guide RNAs were synthesized as oligos (*Supplementary file 1*), annealed, and inserted into the pCasG plasmid using In-Fusion.

The pRSng-CLD-PyrKII plasmid along with the gene-specific pCasG plasmid were co-transfected into the PfMev line, using methods described above. Parasites were selected for 14 days with 1.5 µM DSM1 and 2.5 nM WR99210, after which they were cultured with CMA medium. The parasites were then single-cloned via limiting dilution. The insertion of the CLD was validated using the primers listed in *Supplementary file 1*, screening for insertion as well as the potential presence of any contaminating wild-type parasites. Primers were designed to screen for 5' integration (Δ5' reaction primers PyrK.CLD.5.F and CLD.R) and 3' integration (Δ3' reaction primers PyrK.CLD.3.R and CLD.F) of the CLD, and the 5' region (primers PyrK.CLD.5.F and PyrK. CLD.WT5.R) and 3' region (primers

PyrK.CLD.WT3.F and PyrK.CLD.3.R) of the *pyrKII* gene. The parental PfMev line was used as a control for these reactions.

## TetR-DOZI PyrKII and CLD-PyrKII growth curves

Parasite lines were synchronized via magnetic purification (*Mata-Cantero et al., 2014*), by passage through a MACS LS column (Miltenyi Biotec) placed in a custom made magnet with a field strength of about 8000G 6 hr prior to initiating growth experiments. TetR-DOZI PyrKII parasites were washed three times with 10 mL CMA to remove aTc and then seeded at 0.5% starting parasitemia and 2% hematocrit in a 96-well plate. Quadruplicate cultures were set up with and without 500 nM aTc. Parasite samples were collected for quantification by flow cytometry (*vide ante*) at 16 hr and daily thereafter with the culture media exchanged at those time points. On day 4, parasites were cut 1:10. The growth curve for the CLD-PyrKII line was conducted similarly, with the exception that the parental line was not washed, as there was no need to remove any ligand. Quadruplicate cultures were set up with and without 500 nM Shield1. Data from two biological replicates for each line were analyzed using two-way ANOVA in Prism (GraphPad).

## Tracking apicoplast loss via live epifluorescence microscopy

The CLD-PyrKII parasite line was synchronized via magnetic purification and used to seed a 50-mL culture flask at 1% parasitemia and 2% hematocrit. Six hours post-synchronization (mid-schizogeny) 500 nM Shield1 was added, representing time 0. Media containing 500 nM Shield1 were replaced at 16 hr and every 24 hr thereafter. Samples were collected at time points 0, 16, 40, 64, 88, and 110 hr and the morphology of the apicoplast was assessed via live epifluorescence microscopy. For each time point, parasites were collected and processed using the same protocol for live epifluorescence microscopy as described above. Apicoplast morphology was assessed by visual inspection of at least 15 parasites from each time point to determine whether the apicoplast was intact or disrupted. Average data are presented from three independent biological replicates.

## Transcriptomic analysis of CLD-PyrKII parasites

Samples of the CLD-PyrKII parasite cultures described above were also prepared at 8 hr intervals for transcriptomic analysis. For each time point, 5 mL of parasite culture was centrifuged at 400 g for 5 min, and 100 μL of the pelleted cells were snap frozen with dry ice and stored at −80℃. RNA was extracted from 100 μL of packed RBC culture with Trizol Reagent and the PureLink Mini Kit (Invitrogen/Thermo Fisher Scientific). 1 mL Trizol was added to each sample, followed by high-speed disruption in tubes containing Lysing Matrix D (MP Biomedical) in a FastPrep 120 Instrument at speed 6 for 20 s. Homogenates were subsequently processed according to the PureLink Mini Kit manufacturer's protocol (Invitrogen/Thermo Fisher Scientific), including on-column DNase treatment. Following elution of purified RNA from the PureLink columns with Nuclease-free water, quantitation was performed using a NanoDrop spectrophotometer and quality assessment determined by RNA ScreenTape analysis on an Agilent TapeStation 2200.

From 100 ng of total RNA, Cyanine-3 labeled cRNA was prepared using the Low Input Quick Amp Labeling Kit One-Color (Agilent). The eight time course samples were hybridized (along with the manufacturer's RNA spike-in controls) according to standard protocol to the 8 sectors of an Agilent 8 × 15K platform microarray (AMADID 037237) (*Kafsack et al., 2012*) with separate microarray slides used for each biological replicate (in triplicate for each condition). Post-hybridization washes were performed, followed by scanning with the Agilent G2600D SureScan microarray scanner using scan protocol AgilentHD_GX_1color. Scan Control software automatically performed autofocus, autogain, and gal file download. Agilent's Feature Extraction Software was used to assign grids, provide raw image files per array, and generate QC metric reports from the microarray scan data. The QC metric reports were used for quality assessment of all hybridizations and scans.

Txt files from the Feature Extraction software were imported into the Partek Genomics Suite (v7.0; Partek) for detailed analyses of gene expression. Within Partek, the gProcessedSignal was imported, and the intensity values were normalized to the 75th percentile, lower expressed genes were filtered out using a max cutoff of <= 0.5, and log transformation base 2.0 was performed. A two-way ANOVA with linear contrasts for treatment (PyrKII CLD) and time (8–56 hr) versus an untreated control was performed with outputs of p value, fold change, and mean ratio. Microarray

data have been deposited to the National Center for Biotechnology Information Gene Expression Omnibus repository [GSE136169].

Microarray data were used to compare the average of all apicoplast or nuclear and mitochondrial RNA levels (above the threshold of detection) in the CLD-PyrKII parasite line treated with Shield1 as compared to an untreated control over the course of 56 hr. The RNA levels of the Shield1-treated CLD-PyrKII parasite line were plotted as the fold difference compared to an untreated control. The processed microarray data and calculations are shown in *Supplementary file 2*.

### RT-qPCR of apicoplast and nuclear transcripts inPfMev Δ*dxs* parasites

PfMev Δ*dxs* parasites were synchronized and washed three times to remove mevalonate from the culture. Parasites were then seeded at 0.8% parasitemia in 2% hematocrit cultures and grown in the presence or absence of mevalonate. Samples of 4 mL of culture were collected every 8 hr for 48 hr, snap frozen with dry ice and stored at −80°C. Total RNA was extracted by adding 1 mL Trizol reagent to each sample followed by disruption in tubes containing Lysing Matrix D as described above. Homogenates were further processed using the RNeasy Plus Mini kit (Qiagen) according to the manufacturer's instructions. From 350 ng of total RNA, cDNA was generated using the RevertAid First Strand cDNA Synthesis kit (Thermo Fisher). RT-qPCR was used to assess the abundance of one nuclear transcript, ldh (lactate dehydrogenase, PF3D7_1324900), and three apicoplast genome transcripts: EF-Tu (elongation factor Tu, PFC10_API0028), LSU (large subunit ribosomal RNA, PFC10_API0010) and ClpM (chaperone protein ClpM, PFC10_API0060). StepOne Real-Time PCR machines (Thermo Fisher) were used for quantification of cDNA using gene specific primers (*Supplementary file 1*) and the Power SYBR Green PCR Master Mix (Life Technologies 4367659). Serial dilutions of pure parasite DNA were used as standards for relative quantification. All samples were analyzed in triplicate in independent RT-qPCR experiments. The primer sequences used to amplify EF-Tu were taken from *Briolant et al., 2010*.

### Change in luciferase-mediated luminescence in the apicoplast upon mislocalization of PyrKII

The p15-Mev-aSFG plasmid (*Swift et al., 2020*) was modified to express an apicoplast-localized luciferase in the PfMev CLD-PyrKII parasite line. To accomplish this, plasmid p15-p230-aFluc-mCh was constructed from plasmid p15-Mev-aSFG (*Swift et al., 2020*). This plasmid was digested with AvrII and AflII and a construct including firefly luciferase was inserted. The construct encodes the first 55 amino acids of ACP for localization to the apicoplast organelle, firefly luciferase containing the K549E mutation to limit peroxisomal trafficking, and a localization tag composed of a hemagglutinin tag (HA) appended to the red fluorescent protein mCherry. This protein-coding construct was then followed by a 10x aptamer sequence for tetR-DOZI knockdown (*Ganesan et al., 2016*). In this case, however, the 860 bp aptamer array found in pMG74 (*Ganesan et al., 2016*) was replaced with a 619 bp 10x aptamer array designed to be less repetitive and avoid issues with aptamer recombination (*Rajaram et al., 2020*). The sequence of the final construct including the new aptamers is shown in *Figure 6—figure supplement 4*. BamHI and HindIII were used to remove the mevalonate gene from p15-Mev-aSFG and insert the tetR-DOZI coding region, a p2A skip peptide and BSD, as found in pMG74 (*Ganesan et al., 2016*). We then added homology arms (HA) to facilitate site-specific integration into the *p230p* gene (PF3D7_0208900), which is dispensable in blood stage parasites (*Marin-Mogollon et al., 2018*). HA1 and HA2 were sewn together with PCR primers p230HA1.F and p230HA1.NotI.R (for HA1) and primers p230HA2.NotI.F and p230HA2.R (for HA2) and inserted into the NotI site in the backbone of p15-Mev-aSFG (*Supplementary file 1*). Primers p230HA1.F and p230HA2.R introduce SalI sites between HA1 and HA2, allowing the plasmid to be linearized prior to transfection (*Figure 6—figure supplement 3*). Guide RNA primers p230p.gRNA.F and p230p. gRNA.R were annealed and inserted into pCasG (*Swift et al., 2020*; *Rajaram et al., 2020*) as described above in the methods for gene knockout plasmids.

SalI-linearized p15-p230-aFluc-mCh plasmid along with the gene-specific pCasG plasmid were co-transfected into the PfMev CLD-PyrKII parasite line, using methods described above. Parasites were selected for 7 days with 1.5 µM DSM1 and 2.5 µg/mL blasticidin, after which they were cultured with CMA medium. The parasites were then single-cloned via limiting dilution. Integration of p15-p230-aFluc-mCh into the P230p locus was validated using the primers listed in

*Supplementary file 1*. Primers were designed to screen for 5' integration (Δ5' reaction primers p230up.F and p15int.R) and 3' integration (Δ3' reaction primers p15int.F and p230down.R). Primers p230up.F and p230down.R were used to detect the parental PfMev CLD-PyrKII line (*Figure 6—figure supplement 3*). Live epifluorescence microscopy was conducted as described above except that MitoTracker was not used in order to observe native mCherry fluorescence. Red fluorescence was not observed unless the 500 nM aTc was added to enable proper gene expression (*Figure 6—figure supplement 3*).

The PfMev CLD-PyrKII api-luciferase line was used to assess relative ATP levels in the apicoplast organelle. Parasites maintained in 500 nM aTc were synchronized via magnetic purification and used to seed a 15 mL culture flask at 1% parasitemia and 2% hematocrit. Six hours post synchronization (mid-schizogeny) the culture was split into four 3.5 mL fractions and the media were replaced with different growth media, all of which contained 500 nM aTc: CMA, Glucose-free CMA (made with US Bio R9010), CMA with 500 nM Shield1, and CMA with 50 μM fosmidomycin. None of these conditions contained mevalonate. Triplicate 1 mL cultures were set up for each condition and luminescence measurements were taken, representing time 0. Luminescence was measured in black 96-well plates (Corning 3792) with 100 μL of parasite culture and 2 μL of 20 mM D-luciferin (PerkinElmer #122799) using an IVIS Spectrum (PerkinElmer). Measurements were made every 8 hr and the culture media were replaced at 16 and 40 hr. Biological duplicates were analyzed for the glucose-free condition while all other conditions were conducted in triplicate (each with technical triplicates).

## Acknowledgements

This work was supported by the National Institutes of Health R01 AI065853 and R21 AI101589, the Johns Hopkins Malaria Research Institute, and the Bloomberg Family Foundation. CK was supported by NIH PREP grant R25GM109441 and KR was supported by NIH training grant T32AI007417.

## Additional information

### Funding

| Funder | Grant reference number | Author |
| --- | --- | --- |
| National Institute of Allergy and Infectious Diseases | R01AI065853 | Sean T Prigge |
| National Institute of Allergy and Infectious Diseases | R21AI101589 | Sean T Prigge |
| National Institute of General Medical Sciences | R25GM109441 | Cyrianne Keutcha |
| National Institute of Allergy and Infectious Diseases | T32AI007417 | Krithika Rajaram |

The funders had no role in study design, data collection and interpretation, or the decision to submit the work for publication.

### Author contributions

Russell P Swift, Conceptualization, Formal analysis, Investigation, Visualization, Methodology; Krithika Rajaram, Resources, Investigation; Cyrianne Keutcha, Formal analysis, Investigation; Hans B Liu, Formal analysis, Investigation, Methodology; Bobby Kwan, Investigation; Amanda Dziedzic, Data curation, Formal analysis, Investigation; Anne E Jedlicka, Data curation, Investigation, Project administration; Sean T Prigge, Conceptualization, Resources, Supervision, Funding acquisition, Investigation, Visualization, Methodology, Project administration

### Author ORCIDs

Krithika Rajaram (iD) http://orcid.org/0000-0003-4830-5471
Sean T Prigge (iD) https://orcid.org/0000-0001-9684-1733

Decision letter and Author response
Decision letter https://doi.org/10.7554/eLife.50807.sa1
Author response https://doi.org/10.7554/eLife.50807.sa2

## Additional files

### Supplementary files
- Supplementary file 1. Oligonucleotides used in this study.
- Supplementary file 2. Analysis of microarray data.
- Transparent reporting form

### Data availability
Microarray data have been deposited in GEO under accession code GSE136688.

The following dataset was generated:

| Author(s) | Year | Dataset title | Dataset URL | Database and Identifier |
|---|---|---|---|---|
| Swift R, Keutcha C, Liu H, Rajaram K, Dziedzic A, Jedlicka A, Prigge S | 2019 | The NTP Generating Activity of Pyruvate Kinase II is Critical for Apicoplast Maintenance in *Plasmodium falciparum* | http://www.ncbi.nlm.nih.gov/geo/query/acc.cgi?acc=GSE136688 | NCBI Gene Expression Omnibus, GSE136688 |

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
