## [Decision Letter]

**Acceptance summary:**

In this study, the authors made use of the resource of an important new parasite line that bypasses the sole metabolic pathway within the Plasmodium plastid (apicoplast) that is essential during asexual culture. This allows them to test which apicoplast metabolic steps are essential, and to dissect whether they are important for metabolic pathways that serve the rest of the parasites, or are instead necessary for apicoplast maintenance. They demonstrate that pyruvate kinase II is essential for organelle maintenance due to its unexpected ability to generate a broad range of nucleotide triphosphates.

**Decision letter after peer review:**

Thank you for submitting your article "The NTP generating activity of pyruvate kinase II is critical for apicoplast maintenance in *Plasmodium falciparum*" for consideration by *eLife*. Your article has been reviewed by three peer reviewers, and the evaluation has been overseen by a Reviewing Editor and Neil Ferguson as the Senior Editor. The reviewers have opted to remain anonymous.

The reviewers have discussed the reviews with one another and the Reviewing Editor has drafted this decision to help you prepare a revised submission.

Summary:

The authors provide a detailed genetic and functional study of six apicoplast proteins with central roles in Plasmodium organelle metabolism. Results for PDH and DXS are consistent with prior inhibitor and/or genetic studies of PDH and the MEP pathway. The important novelties of this study are (1) observation that iTPT (a proposed inner-membrane 3-carbon phosphate transporter) and the apicoplast-targeted pyruvate kinase II are essential for apicoplast biogenesis independent of their previously proposed roles in supporting MEP pathway isoprenoid synthesis and (2) the insightful proposal (with strong though not air-tight support) that PyrKII is essential for producing multiple NTPs needed for broad apicoplast metabolism.

Essential revisions:

1) The authors present microarray data showing diminished apicoplast versus nuclear RNA levels upon PyrKII mislocalization as evidence for a specific role for this enzyme in supporting RNA transcription. This experiment lacks a negative control to rule out that defects in RNA abundance can result non-specifically from any lethal apicoplast dysfunction. For example, is RNA abundance affected upon mevalonate washout from the DXS KO line, since DXS/MEP pathway activity is not expected to directly influence RNA abundance? A full microarray analysis is not necessary, and a simple RT-qPCR of SufB versus nuclear LDH would suffice.

2) Is broad NDP substrate specificity by PyrKII required for its essential role in apicoplast biogenesis apart from pyruvate production? Does a cytosolic PyrK homolog with much narrower substrate specificity rescue loss of the apicoplast PyrKII and its broad(er) NDP substrate specificity? (Suggestion: do an episomal rescue of their conditional mislocalization, where they transfect the CLD line with plasmids constitutively expressing PK variants with differing NTP specificities and ask which rescue(s) effects of mislocalizing the endogenous PKII).

3) It would be the best if authors can directly check the NTP/dNTP levels in apicoplast, by means like biosensors. It will also be important to see the how timing of NTP/dNTP/mRNA level changes correlate with the PyrK2 mislocalization. In addition, to further support the authors' model, would they see the same phenotypes if they CLD tag DXS in WT parasites (or other genes like DMT2 known to affect the apicoplast maintenance but do not affect NTP/dNTP levels)?

4) If PyrKII is essential for broad NTP production, where do the NDP precursors come from and why are they but not NTPs available in the apicoplast? The mechanisms here may be incompletely understood but explicit discussion of this point at the end of the paper seems warranted.

---

## [Author Response]

Essential revisions:1) The authors present microarray data showing diminished apicoplast versus nuclear RNA levels upon PyrKII mislocalization as evidence for a specific role for this enzyme in supporting RNA transcription. This experiment lacks a negative control to rule out that defects in RNA abundance can result non-specifically from any lethal apicoplast dysfunction. For example, is RNA abundance affected upon mevalonate washout from the DXS KO line, since DXS/MEP pathway activity is not expected to directly influence RNA abundance? A full microarray analysis is not necessary, and a simple RT-qPCR of SufB versus nuclear LDH would suffice.

The suggestion here is to use the DXS KO line and observe RNA abundance after removing mevalonate from the growth medium. We conducted this experiment with synchronized ΔDXS parasites and collected parasite samples at 8hr increments. We used three RT-qPCR probes to apicoplast RNAs in addition to the suggested LDH probes. The results are now shown in Figure 6—figure supplement 2 and show that there is no significant decrease in RNA abundance until hour 48 of the experiment. Thus, during most of the first growth cycle, apicoplast RNA levels are independent of isoprenoid synthesis.

2) Is broad NDP substrate specificity by PyrKII required for its essential role in apicoplast biogenesis apart from pyruvate production? Does a cytosolic PyrK homolog with much narrower substrate specificity rescue loss of the apicoplast PyrKII and its broad(er) NDP substrate specificity? (Suggestion: do an episomal rescue of their conditional mislocalization, where they transfect the CLD line with plasmids constitutively expressing PK variants with differing NTP specificities and ask which rescue(s) effects of mislocalizing the endogenous PKII).

The idea here is to provide more evidence that NTP production by PyrKII is essential for apicoplast biogenesis rather than pyruvate production. Earlier in the project we thought a lot about using other PyrK enzymes as probes, but we ran into several conceptual problems. The first has to do with demonstrated substrate specificity. The most homologous probe would be the apicoplast PyrKII from *T. gondii* since it is the only other type II pyruvate kinase that has been studied. While this enzyme has been shown to prefer GDP over ADP, no other nucleotides were tested. We would have to fully characterize recombinant TgPyrKII before we could determine whether it would be an appropriate probe of nucleotide metabolism. Even less is known about the specificity of the cytosolic *P. falciparum* PyrKI enzyme. Many pyruvate kinases from bacteria and other sources have also been studied, but none have been characterized with the full range of possible substrates. Regardless of which pyruvate kinase is chosen as a probe, the real problem has to do with proving that the probe is enzymatically active after expression in the apicoplast. If the probe fails to compensate for the loss of PyrKII, it could be because the probe enzyme has a different substrate specificity, or it could be because it is not active. This is a significant concern because the activity of type I pyruvate kinases is modulated by many factors including the binding of effectors like alanine, fructose 1,6-bisphosphate and potassium ions.

Instead of using a pyruvate kinase as a probe, we tried to express an enzyme that consumes pyruvate, without affecting the nucleotide pool. We chose the well-characterized lactate dehydrogenase (LDH) from *P. falciparum* as the probe with the hypothesis that LDH expression would deplete pyruvate and affect isoprenoid synthesis, but not cause apicoplast loss. Importantly, we had a method to confirm LDH activity in the apicoplast. Parasites expressing active LDH should require mevalonate supplementation to compensate for loss of isoprenoid synthesis.

We expressed LDH with an amino-terminal conditional localization domain (CLD) to control apicoplast localization and appended a carboxy-terminal mCherry protein for verifying localization with live fluorescence. We also included a new 10x aptamer array for an additional level of control using the TetR-DOZI system. We were able to obtain parasite clones with proper insertion of the transfection plasmid (see Author response image 1). When we added anhydrotetracycline (aTC) to induce full production of LDH, we did not observe parasite death (see Author response image 1). Expressed CLD-LDH-mCherry was largely apicoplast localized (M1=0.42, n=17), but often appeared as puncta in the region defined by the apicoplast (see Author response image 1). Under mevalonate supplementation conditions, this result could be interpreted as confirming our hypothesis that pyruvate is not essential for apicoplast biogenesis. However, the parasites also grew without mevalonate. This result showed that the LDH probe did not deplete enough pyruvate to affect isoprenoid synthesis. Since we could not confirm LDH activity, we did not include these results in the paper. This example highlights why we would have had difficulty using another pyruvate kinase as a probe. Without a mechanism to assess the activity of the probe, it would be impossible to draw any conclusions from the experiment.

**Author response image 1. sa2fig1:** Expression and localization of lactate dehydrogenase (LDH) in the apicoplast. A) Genotyping PCR confirming insertion of CLD-LDH-mCherry (clone E7), with amplification demonstrating plasmid integration at the 5’ and 3’ loci of the P230p locus (lanes a, b), and lack of wild-type parasites due to failure to amplify at the wild type locus (lane c), as compared to the parental control (WT). B) Growth curve of the CLD-LDH-mCh parasite line cultured with 500nM aTC (aTC), aTC and 50μM mevalonate (Mev), or without supplements (CMA). C) Live epifluorescence microscopy of the CLD-LDH-mCherry parasite line. This line expresses api-SFG (green) and CLD-LDH-mCherry (red), while DAPI (blue) marks the nucleus. Microscopy images represent fields that are 10μM long by 10μM wide and include Mander’s values and positive product of the differences from the mean (+vePDM) panels.

3) It would be the best if authors can directly check the NTP/dNTP levels in apicoplast, by means like biosensors. It will also be important to see the how timing of NTP/dNTP/mRNA level changes correlate with the PyrK2 mislocalization. In addition, to further support the authors' model, would they see the same phenotypes if they CLD tag DXS in WT parasites (or other genes like DMT2 known to affect the apicoplast maintenance but do not affect NTP/dNTP levels)?

This suggestion is similar to point 2, above, in that the idea is to better connect PyrKII to nucleotide metabolism in the apicoplast. We searched for an appropriate biosensor, but did not find one that would allow us to measure the levels of multiple nucleotides. Instead, we focused on using firefly luciferase to track the abundance of one nucleotide, ATP. We inserted a luciferase expression cassette into the P230p locus in PfMev CLD-PyrKII parasites. After confirming protein expression and localization, we conducted experiments to determine how mislocalization of PyrKII affects ATP levels in the apicoplast. Starting with synchronized parasites, we mislocalized PyrKII and quantified the change in luminescence every 8 hours for 48 hours. We found that mislocalization of PyrKII results in a decrease in the level of luminescence, suggesting that PyrKII may be playing an important role in ATP production/regeneration within the organelle (See Figure 6D in the manuscript). Figure 6—figure supplement 3 and Figure 6—figure supplement 4 provide additional information about this approach.

The second part of this question is similar to the first question, inquiring if the same trends in apicoplast RNA loss are the same between PyrKII and an essential apicoplast protein not required for organelle maintenance, such as DXS. We have conducted the experiment in which we remove mevalonate in DXS KO parasites and have shown that the loss of apicoplast RNA levels does not follow the same trend as that seen when mislocalizing PyrKII. While this experiment is not exactly the same as mislocalizing DXS it does approximate a similar effect that would have likely occur if we had, because mevalonate is bypassing the requirement for DXS, and is then removed. While we recognize that we were not measuring NTP/dNTP levels directly in that experiment, we believe that measuring RNA levels is a fair approximation.

4) If PyrKII is essential for broad NTP production, where do the NDP precursors come from and why are they but not NTPs available in the apicoplast? The mechanisms here may be incompletely understood but explicit discussion of this point at the end of the paper seems warranted.

Unfortunately, there aren’t very many studies that shed light on nucleotide metabolism in the apicoplast. We added a paragraph to the Discussion section describing apicoplast nucleotide metabolism and whether various metabolic steps appear to occur in the organelle.